# PPI Candidate Ranking:
# Large-Scale Evaluation of a Domain Knowledge-Guided Pipeline

## Abstract

Computational approaches have become central to Protein–Protein Interaction (PPI) research, complementing experimental techniques that remain costly and incomplete. While modern deep learning methods capture diverse biological signals and hold promise in expanding the known interactome, empirical validation remains a critical bottleneck due to its long and expensive procedures. To address this challenge, we introduce the problem of PPI candidate ranking, aiming to prioritize interactions for experimental testing. We propose a novel framework that leverages domain knowledge through interpretability-guided ranking and further refines prioritization by integrating complementary sources of evidence, including interaction scores, structural plausibility, and biomedical language features. Evaluations on a large-scale dataset constructed from successive STRING releases demonstrate that our approach yields significant improvements over two state-of-the-art PPI prediction models, providing more accurate and biologically coherent rankings.

## 1 Introduction

Protein-protein interactions (PPIs) are essential to cellular life, as they regulate activity, tune specificity, and create the assemblies that execute cellular functions (Ngounou Wetie et al., 2014). Disruptions of interactions, through mutations, altered expression, or mislocalization, can reshape pathways and give rise to cancer, neurodegeneration, immune disorders, and infections (Tang et al., 2023). Mapping PPIs consequently occupies a key role in systems biology because it allows proteins to be put into a relationships' context that guides drug design and interpretation of genetic variation. Predicting PPIs remains a significant challenge for several reasons. First, experimental techniques such as yeast two-hybrid (Brückner et al., 2009) and mass spectrometry (Back et al., 2003) are costly, time-intensive, and incomplete. As a result, only a limited fraction of the human interactome has been experimentally resolved, creating a large discovery gap. A further issue is data heterogeneity (Huang et al., 2016): PPIs arise from diverse molecular determinants such as sequence homology, structural complementarity, and cellular context, making them difficult to predict using only an isolated source of evidence. Additionally, interaction data is not static: interactions may be condition-specific, transient, or modulated by post-translational events, making large-scale mapping infeasible.

Traditional bioinformatics approaches apply alignment (Altschul et al., 1990; 1997; Söding, 2005), domain co-occurrence (Singhal & Resat, 2007; Zhang et al., 2016), and structure modeling (Dey et al., 2013; Zhang et al., 2011; 2012) to predict PPIs, but generalize poorly outside known families (Lewis et al., 2012). More recently, knowledge graphs have emerged as integrative frameworks that unify proteins, drugs, diseases, and pathways into relational networks. Resources such as BioKG (Walsh et al., 2020) and PPIKG (Xiong et al., 2024) standardize heterogeneous data into learning-ready triples, enabling link prediction with KG embeddings and graph neural networks. Yet, they remain constrained by coverage gaps and tend to underperform on rare or novel proteins. Sequence-based approaches such as D-SCRIPT ~~(Sledzieski et al., 2021b)~~ (Sledzieski et al., 2021a) and Topsy-Turvy (Singh et al., 2022), on which we focus in this work, use protein language embeddings to predict interactions directly from amino acid sequences; however, because they operate only on sequence, they do not explicitly account for tertiary/quaternary structure, conformational

dynamics, or context-dependent binding interfaces. In parallel, structure-based pipelines such as AlphaFold (Jumper et al., 2021) reconstruct 3D dimeric complexes, offering structural models that can be used to validate predicted contacts, but these pipelines remain computationally intensive and are generally too slow to apply at large scale. Despite these advances, it remains uncertain whether computational models can truly extend the interaction landscape of a known protein. Network databases such as STRING (Mering et al., 2003) aggregate heterogeneous evidence into probabilistic interaction maps, but their predictions require confirmation through in vitro experiments. Existing PPI benchmarks are largely static and retrospective, as models are typically evaluated within a single database release. Such evaluations do not assess whether computational methods can anticipate interactions that will only be experimentally confirmed in future updates, leaving the prospective value of current predictors unclear.

In this work, we address this problem by introducing *PPI candidate ranking*: given a target protein and a set of known interacting proteins, we rank novel candidates that are most likely to interact with the target. This task directly responds to the discovery gap by guiding in vitro experiments toward the most promising predictions. Rather than simply approaching PPI prediction as a classification over two proteins, we consider a ranking approach that also takes into account previous knowledge of the target protein. The underlying idea is that novel interactions of a target protein should follow similar mechanisms to already observed interactions. For this reason, we exploit the interpretable structure of D-SCRIPT and Topsy-Turvy by studying the embedding activations that model known interactions to rank the plausibility of novel ones. This framework extends the case-specific strategy of Borghini et al. (2024), originally applied to the NKp46-CALR protein pair (Sen Santara et al., 2023), into a systematic and interpretability-guided approach for prospective PPI prediction across the human interactome. Importantly, we do not frame interpretability here as a means to generate explanations for users; rather, we leverage interpretable model structures as a methodological device to exploit internal representations for ranking.

Building on the assumption that novel interacting proteins are likely to follow patterns similar to those of already discovered ones, we propose a refinement step to enhance ranking quality by integrating biological knowledge about proteins. In particular, compared to Borghini et al. (2024), we also enrich this approach by proposing a re-ranking strategy on the best predicted interactions. The re-ranking incorporates additional signals, including interaction scores, structure-derived features, and semantic or language-model-based evidence. We show that this step is crucial to refine the initial embedding-based ranking and to quantify how complementary evidence improves prioritization.

For a large-scale evaluation, we validate our framework using two consecutive versions of the STRING database. Known interactions from v11 serve as the basis for retrieval, while novel interactions appearing in v12 provide a prospective test set. In this setting, we show that, with respect to predicted interaction scores by ~~DSCRIPT~~ D-SCRIPT and Topsy-Turvy, we improve ranking metrics by two orders of magnitude.

The contributions of our work are the following:

- We introduce the problem of PPI Candidate Ranking;

- We develop a general interpretability-guided framework to assess the prospective value of PPI prediction methods;

- We adapt 4 techniques to refine PPI candidate rankings;

- We perform a large-scale evaluation across STRING v11-v12 transitions, quantifying how embedding activation analyses compare with alternative signals;

- We analyze complementarities between methods, highlighting which features are most effective at anticipating genuine novel interactions;

The remainder of this work is organized as follows. Section 2 reviews related work, and Section 3 provides the necessary background for understanding our approach. Section 4 details the proposed framework, Section 5 presents experiments and results, and Section 6 concludes with implications and future directions.

## 2 RELATED WORK

Here, we present the main literature related to our work. We start by introducing the field of PPI with early approaches. We then follow by presenting sequence-based approaches, that are at the core of our methodology. Finally, we conclude with latest approaches involving 3-D structures and large language models (LLMs), which we use in our work to refine our PPI candidate rankings.

**Early Experimental and Computational Methods.** Early efforts to uncover PPIs rely on experimental assays such as yeast two-hybrid (Y2H) (Brückner et al., 2009), co-immunoprecipitation (Co-IP) (Lin & Lai, 2017), and tandem affinity purification (TAP) (Rigaut et al., 1999). Although the first large-scale datasets are built upon them, they suffer from high false-positive rates and limited coverage of transient interactions. The use of mass spectrometry allows for coverage, however it faces reproducibility and validation issues (Back et al., 2003; Wepf et al., 2009). To complement experimental evidence, computational inference leverages conserved molecular features: structure-based transfer propagates interactions across homologous folds (Dey et al., 2013; Zhang et al., 2011; 2012), while domain–domain models use modular binding units as predictors of molecular recognition (Singhal & Resat, 2007; Zhang et al., 2016). This concept expands into interolog transfer and network-based methods, which apply evolutionary conservation and graph-theoretic signals to detect conserved interactions and disease-associated modules (Lewis et al., 2012; Kim et al., 2022; Rout et al., 2024).

**Learning-based Approaches for PPI.** Machine learning shifts PPI prediction from handcrafted features to data-driven models. Early approaches rely on engineered descriptors (Ye et al., 2023), but struggle with scalability and generalization. Deep Learning overcomes these limits by learning features directly from raw sequences, with embeddings outperforming designed descriptors (Alakus & Turkoglu, 2019). In this context, structure-informed embeddings mark a turning point: Bepler & Berger (2019) propose to train bidirectional LSTMs with SCOP hierarchy and contact map supervision, capturing both global fold information and local contacts. Building on this, D-SCRIPT introduces a contact-map bottleneck that projects residue embeddings into pairwise grids processed by 2D convolutions to predict inter-protein contact probabilities ~~(Sledzieski et al., 2021b)~~ (Sledzieski et al., 2021a). Topsy-Turvy (Singh et al., 2022) extends this paradigm with a diffusion-based loss (GLIDE Devkota et al. (2020)) that integrates network context, improving low-degree and cross-species performance. More recently, xCAPT5 (Dang & Vu, 2024) proposes a hybrid approach that pairs protein language-model embeddings with a neural network and a boosting classifier, capturing interaction signals directly from sequences. In parallel to structure-informed embeddings, 3-D protein structure prediction reshapes the field thanks to the near-experimental accuracy of AlphaFold2 (Jumper et al., 2021). FoldDock (Bryant et al., 2022) adapts AlphaFold2 for heterodimeric complexes via paired MSAs and introduces the pDockQ confidence score, achieving competitive benchmark performance. SpeedPPI further scales this workflow to interactome-level prediction with reduced runtime and storage (Bryant & Noé, 2023). However, compared to sequence-based approaches, structure-based ones require higher computational resources and processing times.

**Language Models and Semantic Similarity.** LLMs emerge as a new paradigm in bioinformatics, driving progress across protein structure prediction, sequence analysis, drug discovery, and biomedical text mining (Sarumi & Heider, 2024). For PPI discovery, protein-oriented models, such as ProteinBERT (Brandes et al., 2022) and ProtGPT-2 (Ferruz et al., 2022), learn directly from amino acid sequences, producing context-aware embeddings that capture evolutionary and structural cues. Text-oriented models, including BioBERT (Lee et al., 2020) and PubMedBERT (Gu et al., 2021), leverage PubMed-scale corpora for relation extraction, entity normalization, and question answering. BioBERT adapts pretrained models to biomedical abstracts, while PubMedBERT trains from scratch on the same corpus; both enabling PPI comparison via textual annotations. More recently, multi-modal scientific models like Galactica (Taylor et al., 2022) integrate sequences, structures, and literature into a unified representation, illustrating the potential of combining molecular and semantic signals. Finally, LLMs can act as re-rankers: BERT-based models (e.g., BioBERT, Blue-BERT) improve candidate prioritization in biomedical text retrieval and normalization tasks (Peng et al., 2019; Cho et al., 2021; Rybinski et al., 2020). Building on these advances, we employ LLMs to refine candidate ranking through protein annotations, adding an interpretable semantic layer that complements sequence- and structure-based evidence.

## 3 BACKGROUND

Many modern PPI prediction models rely on pretrained protein language models as encoders. ~~One of the most widely adopted~~ An example is the Bepler & Berger model (Bepler & Berger, 2019), a bidirectional LSTM trained in a multi-task setting with three complementary objectives: (i) predicting global structural similarity via SCOP (i.e., Structural Classification of Proteins) (Murzin et al., 1995), (ii) predicting residue–residue contacts within proteins, and (iii) aligning homologous sequences through multiple sequence alignments (MSA). The model employs a Soft Symmetric Alignment (SSA) mechanism to compare residue embeddings across sequences, enabling it to capture structural similarity even in cases of low sequence identity. Given an input sequence $S$ of length $n$, the encoder generates residue-level embeddings $E \in \mathbb{R}^{n \times d}$ with $d = 6165$, capturing both local amino-acid context and global evolutionary and structural features. While highly informative, these embeddings are high-dimensional and redundant, requiring further transformation before being used in downstream tasks.

D-SCRIPT is a sequence-to-interaction framework that produces protein representations informed by structural and evolutionary context. The model builds directly on the pretrained Bepler & Berger language embeddings. Its projection module reduces the dimensionality of $E$ through a linear layer followed by ReLU activation and dropout, producing compact latent embeddings. These embeddings are passed through a residue–contact module that predicts a sparse inter-protein contact map, which is subsequently aggregated into an interaction module. For a pair of sequences of length $n$ and $m$, D-SCRIPT produces a residue-residue contact probability matrix $C \in [0,1]^{n \times m}$. Each entry $C_{i,j}$ quantifies the model's estimated probability that residue $i$ of the first protein is in physical contact with residue $j$ of the second protein in a bound complex. In other words, $C$ provides a residue-level structural hypothesis, indicating where potential contacts are most likely between the two proteins. Convolutional and pooling operations then identify consistent local contact patterns and retain the strongest candidate regions. Finally, a logistic activation compresses these features into a single scalar interaction probability $\hat{p} \in [0,1]$, known as the interaction score (IS), which provides a supervised estimate of binding likelihood.

Topsy-Turvy extends the D-SCRIPT framework by integrating top-down supervision from interaction networks. Like D-SCRIPT, it uses the Bepler & Berger encoder followed by a projection module that yields fixed-size activated embeddings, and a convolutional contact module that predicts residue-residue interactions. Its training is augmented with an additional objective: supervision from GLIDE scores derived from the global protein–protein interaction network. This additional loss shapes the latent embeddings to capture not only molecular and structural features but also topological compatibilities derived from the network context. As a result, the Topsy-Turvy latent space encodes both bottom-up biological signals and top-down network-derived interaction priors.

## 4 PPI CANDIDATE RANKING

In this section, we introduce the definition of our problem and outline the overall pipeline.

**Problem Setup.** Let $P$ be a set of proteins and let $I_{\text{known}} \subseteq P \times P$ denote the set of *known interactions* available at a given time. A disjoint set $I_{\text{new}} \subseteq P \times P$ contains *novel interactions* that are discovered only later. Thus, $I_{\text{new}}$ represents interactions that are valid but entirely *unseen* to models trained on $I_{\text{known}}$. For a protein $p \in P$, we define its known partners as

$$KP(p) = \{p_k \in P \mid (p, p_k) \in I_{\text{known}}\}, \tag{1}$$

and its new partners as

$$NP(p) = \{p_n \in P \mid (p, p_n) \in I_{\text{new}}\}. \tag{2}$$

Given $p$ and its known partners $KP(p)$, we construct a ranking $R_p$ over the candidate set $P \setminus KP(p)$, where candidates are ordered by the model's predicted likelihood of interacting with $p$. Evaluation then checks whether the truly novel partners $NP(p)$ appear in high positions within $R_p$.

**Proposed Solution.** To address the candidate ranking problem, we design ~~a~~ the two-stage framework illustrated in Figure 1. First (Section 4.1), we collect the known partners $KP(p)$ of a query protein $p$. These partners are used to model known interactions by identifying how they are modeled in the embedding activations of D-SCRIPT and Topsy-Turvy. These activations are used to produce

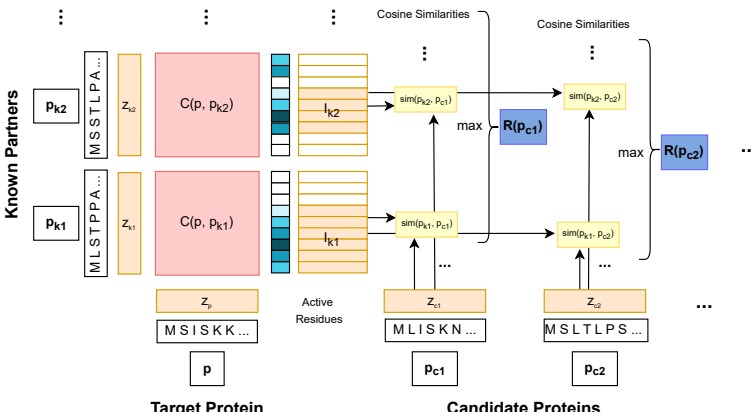

Figure 1: Interpretability-Guided Ranking. Given an input target protein $p$ and a set of known interactors $p_{k1}, ..., p_{kn}$, we compute the predicted contact maps $C(p, p_{k1}), ..., C(p, p_{kn})$. From the maps we extract the indices of the active residues $I_{k1}, ..., I_{kn}$ that are used to select the portion of the embeddings on which we compute the cosine similarity between known partners and candidate properties. Finally, for each candidate, we return the rank score computed as the maximum cosine similarity among all the known interactors.

a first ranking $R_p$ of the candidates by measuring the cosine similarity over activated embeddings. Finally (Section 4.2), the top portion of $R_p$ is refined by incorporating additional biological signals to produce the final ranking.

## 4.1 INTERPRETABILITY-GUIDED RETRIEVAL.

Every protein $p \in P$ is encoded as an embedding $z_p \in \mathbb{R}^{n_p \times d}$ computed from its amino-acid sequence of length $n_p$ using either D-SCRIPT or Topsy-Turvy.

Given a target protein $p$, our aim is to produce a ranking $R_p$ over a set of candidate proteins $CP(p) = \{p_c \mid \forall p_c \in P \setminus KP(p)\}$. To ~~take previous knowledge into account, we employ known interactors $p_k \in KP(p)$ as anchors to define~~ incorporate previous biological knowledge about $p$, we rely on the set of its experimentally supported interaction partners, which we denote as $KP(p)$. These known interactors provide structured prior information on how $p$ typically engages with other proteins and act as anchors for constructing the ranking $R_p$. First, for each $p_k$ we identify the most active region by analyzing the predicted contact map $C(p, p_k) \in [0, 1]^{n_p \times n_{p_k}}$, obtained using D-SCRIPT or Topsy-Turvy. ~~For each residue of~~ The activation score of residue $j$ in $p_k$ ~~, we define its activation score~~ is defined as the maximum contact probability with any residue of $p$, thus capturing the strongest potential interaction signal. ~~The corresponding set of active residue indices is denoted by $I_k \subseteq \{1, ..., n_{p_k}\}$ and contains the contiguous set of residues of~~ We then scan the resulting activation profile along the sequence of $p_k$ and identify all maximal contiguous segments of highly activated residues. Among these segments, we select as $I_k$ the one with the highest average activation score, which we interpret as the most stable and informative interaction region for the pair $(p, p_k)$. We do not impose any fixed window size or number of residues: the length $|I_k|$ is determined by the data and can range from a single residue up to the full sequence of $p_k$ ~~reporting the highest average contact probability.~~.

Successively, we measure the embedding similarity between $p_c$ and every anchor $p_k$, specifically focusing on the region $I_k$. We therefore compute the maximum cosine similarity between the embeddings of active residues of $p_k$, denoted as $z_k[I_k]$ and each consecutive portion of embeddings of $p_c$ of length $|I_k|$:

$$sim(p_c, p_k) = \max_{i=0}^{i < n_c - |I_k|} \frac{\langle z_k[I_k], z_c[i : i + |I_k|]\rangle}{\|z_k[I_k]\|_2 \cdot \|z_c[i : i + |I_k|]\|_2}, \tag{3}$$

where $z_{p_k}$ and $z_{p_c}$ identify the embeddings of $p_k$ and $p_c$, respectively, $z_{p_k}[I_k]$ represent the flattened embeddings of $p_k$ only focusing on the residues in the interval $I_k$, and $z_c[i : i + |I_k|]$ are the flattened embeddings for the residues of $p_c$ between index $i$ and $i + |I_k|$.

The rank score for $p_c$ towards $p$ is then obtained as the maximum similarity over all the anchors $p_k \in KP(p)$:

$$R_p(p_c) = \max_{p_k \in KP(p)} sim(p_c, p_k) \tag{4}$$

Finally, the rankings for the protein $p$ are obtained by sorting (in descending order) all the scores for each $p_c \in CP(p)$:

$$R_p = \text{sort}_{desc\ R_p(p_c)} \{(p_c, R_p(p_c)) \mid p_c \in CP(p)\} \tag{5}$$

## 4.2 RE-RANKING MODULE

Re-ranking operates on candidate sets obtained from the interpretability-guided step. Specifically, due to the heavy processing of some of the techniques, we focus on the top 10 ranked candidates for each target protein $p$. The objective is to incorporate biological signals beyond embedding similarity. To this end, we follow a uniform procedure across multiple signals: for each protein-candidate pair $(p, p_c)$ from the top-$r$ lists, we compute an additional score, re-order candidates accordingly, and evaluate performance in terms of rank shifts. In this sense, a new ranking is obtained for each new signal used. In the remainder of this section, we describe how different signals can be considered.

**Interaction Score.** The first re-ranking signal is the *interaction score* (IS) directly predicted by D-SCRIPT. As explained in Section 3, we use the score $\hat{p} \in [0, 1]$, which provides a supervised estimate of binding likelihood between a candidate pair $(p, p_c)$:

$$\hat{p} = \max_{i \leq n, j \leq m} C(p, p_c)_{ij}, \tag{6}$$

where $C(p, p_c) \in \mathbb{R}^{n \times m}$ is the contact map between $p$ and $p_c$, having sequence lengths of $n$ and $m$ residues, respectively. This probability, sharpened through a logistic activation, is reported as the IS.

**Structural Plausibility via SpeedPPI.** To complement sequence-level and supervised interaction signals, we incorporate a structural view using *SpeedPPI*, an accelerated pipeline built on AlphaFold2. For a candidate pair $(p, p_c)$, SpeedPPI first constructs multiple sequence alignments (MSAs) that are provided to AlphaFold2 where both proteins are folded and docked against each other. Within the re-ranking framework, candidates are primarily ordered by the $p\text{DockQ}$ score from (Bryant et al., 2022), which provides a calibrated estimate of overall complex quality, and it is defined as:

$$p\text{DockQ}(x) = \frac{L}{1 + e^{-k(x - x_0)}} + b, \tag{7}$$

where $x = \text{pLDDT}_{\text{interface}} \cdot \log(\#\text{contacts})$ and $L, k, x_0, b$ are calibration constants estimated from reference complexes. In the rare case of ties, we resolve them by considering secondary indicators: first the average interface interaction confidence from AlphaFold2, and then the raw number of predicted contacts.

**Functional Enrichment & Semantic Scores.** We next add a semantic layer based on functional annotations and free-text summaries. The goal is to test whether *functionally coherent* candidates, measured via ontology terms, domains, pathways, localization, and curated key phrases, receive higher ranks relative to unrelated proteins, showing how semantic evidence helps discriminate plausible partners. Starting from the union of proteins ($p$) and candidates ($p_c$), we map each symbol to a UniProtKB accession and retrieve the following features: gene ontology (GO) terms such as molecular function (MF), biological process (BP), and cellular component (CC); InterPro and Pfam domains, describing conserved families and motifs; Reactome pathways, capturing curated molecular cascades; ComplexPortal complexes, defining experimentally supported assemblies; and Sub-cellular localization notes, from structured UniProt fields and free-text comments. The result is a per-protein record, from which we compose a *text profile* by concatenating all available fields. From

this profile, we then extract three feature sets: (a) the token set $T(p)$, containing all unique words remaining after preprocessing; (b) the localization set $L(p)$, collecting terms matched from a list of subcellular compartments (e.g., nucleus, cytosol); (c) the key-term set $K(p)$, with role words and structural motifs (e.g., receptor, kinase, transmembrane). Within each group defined by a fixed $p$, and for each candidate $p_c$, we compute TF–IDF cosine similarity on text profiles and Jaccard similarity on $T(p)$, $L(p)$, and $K(p)$, measuring the fraction of shared terms between sets. The TF–IDF cosine similarity is defined as:

$$s_{\text{tfidf}}(p, p_c) \;=\; \cos\big(v(p),\, v(p_c)\big). \tag{8}$$

The Jaccard similarity for any two sets $A$ and $B$ is: $J(A, B) \;=\; \frac{|A \cap B|}{|A \cup B|}$, applying it to our previously defined feature sets, this yields the following:

$$s_{\text{overlap}}(p, p_c) = J(T(p), T(p_c)), \quad s_{\text{loc}}(p, p_c) = J(L(p), L(p_c)), \quad s_{\text{key}}(p, p_c) = J(K(p), K(p_c)). \tag{9}$$

Each similarity view produces a within-$p$ ranking, showing how structured annotations and key terms promote coherent candidates:

$$\text{rank}_v(p, p_c) = \text{rank}\big(\{s_v(p, \cdot)\},\ \text{descending}\big), \tag{10}$$

**Large Language Model–based Re-Ranking.** Finally, we test whether LLMs trained on biomedical text can capture functional semantics that go beyond curated ontologies or token overlap. Unlike the semantic scores above, LLMs encode contextual meaning from raw biomedical text, enabling them to detect functional relationships even when terms differ or annotations are incomplete. We consider two approaches: a *bi-encoder* for semantic similarity and a *cross-encoder* fine-tuned for direct ranking. For the bi-encoder setup, sentence-level biomedical encoders (e.g., *Sentence-BioBERT*, *BioMedRoBERTa*) generate fixed-length semantic representations of the text profiles defined above. Each profile is mapped to a dense vector $\phi(x)$, normalized to unit length and candidates are re-ranked by cosine similarity $s_{\text{LLM}}(p, p_c)$:

$$s_{\text{LLM}}(p, p_c) = \cos\big(\phi(p),\, \phi(p_c)\big), \tag{11}$$

where $\phi(\cdot)$ denotes the encoder representation. Within each candidate set, proteins are re-ranked by $s_{\text{LLM}}$, and performance is assessed by changes in the relative position of protein candidates compared to baseline orderings.

To move beyond independent embeddings, we also fine-tune a cross-encoder (*PubMedBERT* backbone) on query–candidate pairs. The query consists of the annotation profile of $p$, while the candidate encodes the profile of $p_c$. Labels indicate whether $p_c \in NP(p)$. Training pairs are constructed exclusively from STRING v11 interactions, and we apply a GroupKFold split by protein identity, ensuring that all examples involving the same protein appear in the same fold. This prevents any protein from occurring in both training and validation sets and eliminates protein-level leakage. All evaluation is performed on novel interactions from STRING v12, which are entirely disjoint from the data used for fine-tuning. The cross-encoder jointly attends to both inputs and outputs a logit $\hat{s}(p, p_c)$ estimating interaction likelihood. At inference, logits are normalized by a within-$p$ softmax:

$$P(p_c \mid p) = \frac{\exp(\hat{s}(p, p_c))}{\sum_{p'_c} \exp(\hat{s}(p, p'_c))}, \tag{12}$$

yielding relative probabilities for ranking. This approach captures contextual functional relationships that structured similarity measures may overlook.

## 5 RESULTS

### 5.1 DATA RETRIEVAL & PREPROCESSING

To ensure compatibility with previous works, we adopt the same preprocessing protocol used in D-SCRIPT and Topsy-Turvy. While those methods are originally developed on STRING v11, we

Table 1: Retrieval performance at different cutoffs $k$, comparing our interpretability-guided method exploiting active embedding regions against direct use of interaction probabilities from D-SCRIPT, Topsy-Turvy and xCAPT5 on STRING v11→v12 novel-partner retrieval.

| | $k$ | Recall ↑ | Precision ↑ | MAP ↑ | nDCG ↑ | Success ↑ | Pred. Cov. ↑ | MRR ↑ | Avg. Rank ↓ | Model |
|---|---|---|---|---|---|---|---|---|---|---|
| **Prediction Probability** | 5 | 0.0071 | 0.0080 | 0.0103 | 0.0098 | 0.0000 | | | | D-SCRIPT |
| | 10 | 0.0124 | 0.0058 | 0.0133 | 0.0100 | 0.0040 | | | | |
| | 50 | 0.0718 | 0.0074 | 0.0718 | 0.0298 | 0.0200 | 0.9544 | 0.0340 | 482.86 | |
| | 100 | 0.1531 | 0.0073 | 0.1531 | 0.0489 | 0.0639 | | | | |
| | 200 | 0.3041 | 0.0057 | 0.3041 | 0.0792 | 0.1617 | | | | |
| | 500 | 0.5855 | 0.0042 | 0.5855 | 0.1298 | 0.4012 | | | | |
| | 5 | 0.0063 | 0.0069 | 0.0100 | 0.0073 | 0.0038 | | | | Topsy-Turvy |
| | 10 | 0.00117 | 0.0077 | 0.0140 | 0.0094 | 0.0058 | | | | |
| | 50 | 0.0639 | 0.0067 | 0.0639 | 0.0248 | 0.0212 | **0.9683** | 0.0256 | 570.52 | |
| | 100 | 0.1492 | 0.0065 | 0.1492 | 0.0446 | 0.0692 | | | | |
| | 200 | 0.2568 | 0.0052 | 0.2568 | 0.0680 | 0.1153 | | | | |
| | 500 | 0.4828 | 0.0038 | 0.4829 | 0.1093 | 0.2827 | | | | |
| | 5 | 0.0452 | 0.1943 | 0.1481 | 0.2000 | 0.0059 | | | | xCAPT5 |
| | 10 | 0.0747 | 0.1848 | 0.1333 | 0.1993 | 0.0069 | | | | |
| | 50 | 0.2285 | 0.1427 | 0.1159 | 0.2202 | 0.0129 | 0.8088 | 0.0315 | 900.11 | |
| | 100 | 0.3177 | 0.0996 | 0.1103 | 0.2416 | 0.1257 | | | | |
| | 200 | 0.4153 | 0.0649 | 0.1162 | 0.2775 | 0.2386 | | | | |
| | 500 | 0.5554 | 0.0341 | 0.1229 | 0.3247 | 0.3603 | | | | |
| **Our Approach** | 5 | 0.1832 | 0.1924 | 0.2714 | 0.2067 | 0.0778 | | | | D-SCRIPT |
| | 10 | 0.2641 | 0.1377 | 0.2952 | 0.2130 | 0.1277 | | | | |
| | 50 | 0.4693 | 0.0437 | 0.4693 | 0.2594 | 0.2715 | 0.9230 | **0.1685** | 239.77 | |
| | 100 | 0.5960 | 0.0263 | 0.5960 | 0.2884 | 0.3753 | | | | |
| | 200 | 0.7193 | 0.0150 | 0.7193 | 0.3122 | 0.5010 | | | | |
| | 500 | 0.8141 | 0.0066 | 0.8141 | 0.3288 | 0.5988 | | | | |
| | 5 | 0.0562 | 0.0586 | 0.0832 | 0.0612 | 0.0244 | | | | Topsy-Turvy |
| | 10 | 0.1106 | 0.0517 | 0.1214 | 0.0765 | 0.0508 | | | | |
| | 50 | 0.3422 | 0.0300 | 0.3422 | 0.1435 | 0.1786 | 0.9506 | 0.0925 | **235.56** | |
| | 100 | 0.4848 | 0.0208 | 0.4848 | 0.1780 | 0.2744 | | | | |
| | 200 | 0.6260 | 0.0114 | 0.6260 | 0.2060 | 0.3947 | | | | |
| | 500 | 0.8105 | 0.0053 | 0.8105 | 0.2367 | 0.5996 | | | | |

apply the identical procedure to the most recent release of the human subset of STRING (v12) (Mering et al., 2003). This guarantees methodological alignment while taking advantage of the expanded interaction space of the new release. To construct a set of high-confidence physical interactions, we retain only binding interactions with experimental support $> 0$, discarding indirect associations (e.g., co-expression, homology, text mining). Protein sequences are filtered to lengths between 50 and 800 residues, to exclude fragments unlikely to fold and ensure computational tractability during embedding generation. To reduce redundancy, sequences are clustered with CD-HIT at a 40% identity threshold, and interactions are removed if both partners collapse into existing clusters. Finally, negative examples are generated by randomly pairing proteins from the filtered, non-redundant set, ensuring no overlap with positives. To reflect the sparsity of interactions in the human proteome, we fix the negative-to-positive ratio at 10:1, labeling positives as 1 and negatives as 0. This procedure yields 279,568 additional positives in v12. The increase from the previous release reflects newly introduced high-confidence binding interactions, driven by high-throughput experiments and structure-based predictions Szklarczyk et al. (2023), which we exploit as ground-truth to assess interpretability-guided retrieval from the embedding space.

## 5.2 EVALUATION METRICS

The performance of retrieval is assessed using standard ranking and recommendation metrics. For each protein, candidate proteins are ranked and the position of the true novel partner(s) is recorded. We report the following ranking-based metrics, each computed at cutoffs $k \in \{5, 10, 50, 100, 200, 500\}$:

- **Recall@k:** Fraction of true partners appearing within the top-$k$ retrieved candidates
- **Precision@k:** Fraction of the top-$k$ candidates that are true partners
- **MAP@k:** Mean Average Precision across proteins, combining precision and ranking quality

Table 2: Pairwise comparison of rank-shifts across evidence sources. For each row–column pair, ~~↑ reports~~ the fraction of STRING v12 interactions whose ranking was maintained or improved when switching from the row method to the column method ~~, and ↓~~ is reported. The background is green if more than an half of ~~the fraction that worsened~~ rankings of the STRING v12 interactions improved, is red otherwise.

| | Cosine | IS | pDockQ | TF-IDF | Token | Location | KeyTerm | BioBERT | BioMed-RoBERTa | PubMed-BERT |
|---|---|---|---|---|---|---|---|---|---|---|
| Cosine | — | 63.0 | 47.2 | 62.4 | 57.9 | 64.8 | 69.3 | 55.8 | 56.1 | 75.5 |
| IS | 48.1 | — | 40.9 | 57.6 | 56.7 | 60.0 | 66.9 | 50.4 | 53.4 | 71.9 |
| pDockQ | 63.6 | 70.7 | — | 69.9 | 66.0 | 71.9 | 75.8 | 67.8 | 62.1 | 79.1 |
| TF-IDF | 52.8 | 62.7 | 39.4 | — | 68.4 | 66.3 | 70.1 | 56.1 | 55.2 | 79.1 |
| Token | 56.1 | 63.6 | 42.7 | 74.0 | — | 69.0 | 75.2 | 57.9 | 56.1 | 78.5 |
| Location | 51.0 | 57.3 | 38.2 | 60.3 | 59.1 | | 69.0 | 48.4 | 46.3 | 69.0 |
| KeyTerm | 45.4 | 53.7 | 33.1 | 52.2 | 53.4 | 56.1 | — | 46.3 | 44.5 | 64.8 |
| BioBERT | 57.9 | 63.0 | 44.2 | 71.6 | 67.8 | 72.8 | 72.5 | — | 65.1 | 79.7 |
| BioMedRoBERTa | 57.6 | 64.5 | 49.0 | 67.2 | 64.5 | 71.3 | 74.0 | 66.0 | — | 79.4 |
| PubMedBERT | 40.9 | 48.1 | 30.4 | 46.3 | 44.2 | 51.9 | 59.1 | 39.4 | 39.7 | — |

- **nDCG@k:** Normalized Discounted Cumulative Gain, rewarding correct hits at higher ranks
- **Success@k:** Fraction of proteins where at least one true partner was found within the top-$k$
- **Prediction Coverage:** Total number of true novel partners that are successfully retrieved across all proteins
- **MRR:** Mean Reciprocal Rank, summarizing early ranking performance
- **Average Rank:** Mean position in the ranking at which true novel partners appear

To further evaluate the effectiveness of the *re-ranking strategies*, we compare baseline similarity retrieval with enriched scorers. Specifically, for each protein we aggregate the top-$r$ candidate lists of its known partners ($r = 10$) using a max-similarity strategy, thereby collapsing duplicates and retaining the highest score observed across partners. This procedure yields 2,280 protein-candidate pairs available for re-ranking analysis. Within this set, we track whether each rediscovered protein *maintaines or improves* its position in the ranking compared to the baseline, or instead *worsens* after re-ranking.

## 5.3 EXPERIMENTS AND RESULTS

We evaluate the effectiveness of our approach in rediscovering novel STRING v12 interactions from v11-derived candidate sets, in two stages: (i) retrieval performance at different cutoffs, and (ii) refinement through re-ranking via pairwise rank-shift analysis. Details of experimental setup and parameter choices are reported in Appendix A.1.

Table 1 compares our interpretability-guided retrieval with direct interaction probabilities from D-SCRIPT~~and~~, Topsy-Turvy ~~. Both baselines recover~~ and xCAPT5. Across all baselines, many novel v12 partners ~~, with~~ are recovered, although the depth at which these rediscoveries appear varies widely. Topsy-Turvy ~~achieving the broadest~~ achieves the broadest prediction coverage due to its network-based design, whereas xCAPT5 shows strong precision in the very early ranks but rapidly decays as $k$ increases, indicating that its probability estimates capture highly confident signals but do not generalize well across the larger candidate space. However, ~~these rediscoveries~~ probability-based rediscoveries from all three models often appear deep in the ranking, limiting their practical value. By contrast, our method substantially reshapes the ranking: for D-SCRIPT, for instance, Recall@10 rises from below 2% to above 25%, and MRR increases by 4-6 times. In absolute terms, this means that out of the first ten suggested proteins, on average more than one in ten (13%) is a true novel partner, an encouraging hit rate for practical candidate screening. This demonstrates how simply using model probability does not represent an optimal choice for modeling the plausibility of novel interactions. On the contrary, exploiting active embedding regions repositions rediscoveries where they matter most for candidate screening. With both approaches, retrieval remains the computational bottleneck, with runtimes in the order of hundreds of hours (Figure 2). At a global level, Topsy-Turvy achieves better performances compared to D-SCRIPT and xCAPT5,

reporting a better coverage and average rank. However, D-SCRIPT appears better suited when focusing on early ranking performances. Indeed, unlike Topsy-Turvy, which disperses rediscoveries broadly but lower in the list, D-SCRIPT emphasizes true partners at the top ranks, yielding higher Precision, MAP, nDCG, and Success. While both converge in large-$k$ coverage, only D-SCRIPT ensures a meaningful subset appears early. For this reason, we select D-SCRIPT as backbone for the re-ranking strategies.

Table 2 presents the pairwise rank-shift analysis of different re-ranking strategies. In particular, we report the fraction of new interactions whose ranking was maintained, improved or worsened when switching from one method to another. Overall, PubMedBERT provides the most consistent positive signal, improving or maintaining 75.5% of rediscoveries and outperforming nearly all other models (e.g., 79.7% vs. BioBERT). D-SCRIPT's interaction score (IS) also adds complementary value, improving 63.0% of rediscoveries compared to cosine, suggesting it is more effective as a refinement signal than as a standalone retrieval criterion. By contrast, pDockQ underperforms for early ranking (47.2% improvement rate), indicating that structural plausibility is better suited for filtering than for direct ordering, a view consistent with its high computational cost compared to the other re-ranking methods (Figure 3). This outcome could be probably justified by the high sensitivity of AlphaFold2 with respect to the seed, when predicting protein interactions. Lightweight heuristics such as TF-IDF and overlaps of Token, Location, and KeyTerm sets yield surprisingly robust improvements (up to ~70% maintain-or-improve rates), showing that even coarse annotation signals can sharpen retrieval lists. This emphasizes the need for including information about the proteins when predicting interactions. Naively, this could be due to the fact that by simply modeling sequences we neglect information about the proteins that is highly important. For instance, it could happen that in theory two proteins could bind but such interaction does not happen due to the scarce probability of the proteins to be in the same environment. BioBERT and BioMedRoBERTa also achieve solid gains (65-72%), but remain slightly below PubMedBERT. Since these models are pretrained on large biomedical corpora, it is uncertain if their gains reflect not only semantic generalization but also latent knowledge of interactions from the training data. Overall, these results highlight the strength of semantic signals, particularly protein descriptions and functional annotations, in capturing relationships missed by sequence- or structure-based methods.

# 6 CONCLUSIONS

In this work, we addressed the challenge of anticipating novel protein-protein interactions emerging in STRING v12 dataset using only information available in the previous version (i.e. STRING v11). We introduced an interpretability-guided retrieval strategy that leverages active embedding regions associated with known partners computed using D-SCRIPT and Topsy-Turvy models. By focusing similarity computations on residues highlighted by predicted contact maps, our method repositions true novel partners higher in the ranked lists.

Our results demonstrate that integrating interpretability-guided retrieval with multi-source re-ranking yields a step change in candidate prioritization, improving early ranking performance by up to two orders of magnitude over existing models. This substantial leap has direct practical consequences: by bringing true novel interactions to the top of the candidate list, our framework enables experimentalists to focus resources on the most promising hypotheses, accelerating discovery and reducing costs.

Despite the promising results, our approach is still subject to limitations. First, it fundamentally relies on the assumption that modeling known interactions of a protein allows for better ranking of its novel interactions. While this assumption yields better results compared to other PPI prediction, it may not hold for underexplored proteins with very few or no known partners. In such cases, the method cannot benefit from case-specific knowledge and reduces to relying solely on generic PPI prediction probabilities. Second, although we exploit interpretability as a structural property of the underlying models to improve ranking, we do not use it as a means of generating explanations. Consequently, the returned rankings remain non-interpretable in the same sense as classical PPI prediction methods. While our framework can identify the residues driving similarity computations, the embedding construction process itself remains a black-box representation. As such, we cannot directly ground the predicted interactions in specific, biologically meaningful protein properties.

Future work will explore such limitations, by including structural priors, extending region-guided similarity to multi-protein contexts, and assessing robustness across diverse organisms for further improving protein-protein interactions discovery.

## 7 REPRODUCIBILITY STATEMENT

All the code needed to run and reproduce the experiments is available in the supplementary material attached to this paper. After acceptance, the material will be published on a public repository.

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

# A  APPENDIX

Table 3: Example annotation record for a sample protein (AAMP)

| Field | Example content (AAMP) |
|---|---|
| Name | AAMP |
| STRING_ID | 9606.ENSP00000403343 |
| Alias | AAMP |
| Annotation | Angio-associated migratory cell protein; plays a role in angiogenesis and cell migration. |
| UniProt accession | Q13685 |
| GO_MF | heparin binding |
| GO_BP | angiogenesis; cell differentiation; positive regulation of endothelial cell migration; smooth muscle cell migration |
| GO_CC | cell surface; cytosol; intercellular bridge; microtubule cytoskeleton; plasma membrane |
| Domains | WD40 repeat domains (e.g., IPR001680, PF00400) |
| Pathways | VEGF signaling; EGFR signaling; Thromboxane signaling; NOD1/2 signaling |
| Complexes | — |
| Notes | Cell membrane; Cytoplasm |

## A.1  MODEL CONFIGURATION AND HYPERPARAMETER SELECTION

All experiments are executed on an NVIDIA Tesla V100 GPU with CUDA-enabled PyTorch. We consider three complementary models in our setup.

The first component is D-SCRIPT, for which we use the official human pretrained model. It produces residue-level embeddings from the Bepler & Berger encoder, pooled into fixed-length vectors. Evaluation is restricted to pairs with predicted Interaction Score $\geq 0.5$. For Topsy-Turvy, we use the official pretrained model with the same hyperparameters as D-SCRIPT, ensuring a fully consistent configuration across both methods.

For SpeedPPI, we assess structural plausibility using FoldDock-based complex modeling combined with the pDockQ scoring function. Multiple sequence alignments are built with `hhblits` against UniClust30, and large-scale parallelism is used to handle candidate batches efficiently. We summarize the configuration details below:

- MSA generation: `hhblits` (UniClust30, $E \leq 10^{-3}$, 4 CPUs/job)

- Parallel jobs: 72 (across 80 CPU cores)

- AlphaFold recycles: 3 (reduced to 0 if combined sequence length $> 1300$ residues)

- pDockQ threshold: 0.0, to retain all predictions and enable analysis of ranking shifts

- GPU memory: growth enabled, preallocation disabled

Finally, we perform semantic re-ranking with an LLM-based CrossEncoder initialized from Pub-MedBERT. Training pairs are constructed from v11 partners, with negatives downsampled to maintain balance. Validation uses a GroupKFold split by protein identity. Its main training parameters were:

- Batch size: 16

- Learning rate: $1 \times 10^{-5}$ (AdamW optimizer)

- Sequence length: max 512 tokens

- Negative sampling: up to 3 negatives per positive

- Precision: FP32 (AMP disabled for stability)

- Inference: batch size 32, logits normalized per-$p_1$ by softmax

## A.2 PERFORMANCE ANALYSIS

By analyzing runtime, we can better appreciate the trade-offs between retrieval and re-ranking.

Both the prediction-probability baseline and our interpretability-guided method require evaluating each known partner against the entire protein set, making retrieval inherently expensive, with runtimes in the order of hundreds of hours. This illustrates that retrieval is the computational bottleneck of the pipeline, thus dominating the overall runtime. The values in Figure 2 are averages across D-SCRIPT and Topsy-Turvy implementations, meaning each method was executed with both models and the mean runtime is reported rather than their sum.

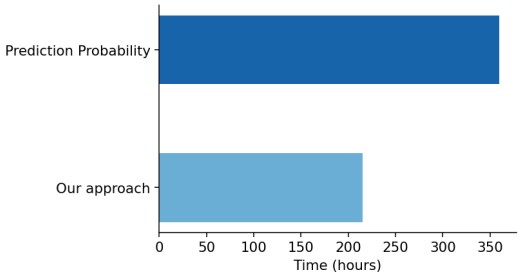

Figure 2: Retrieval runtimes for the prediction-probability baseline and our approach.

Re-ranking has been evaluated on a restricted set of 2,280 pairs (top-10 for each query). The collected runtime reflects only the computation of semantic similarity scores, assuming that all functional annotations are already available locally. The preprocessing or data-retrieval time required to fetch such annotations is not included, and this condition is applied uniformly across all semantic methods to ensure a fair comparison focused solely on the intrinsic algorithmic cost of each re-ranking strategy. Figure 3 highlights substantial differences in runtime across methods. SpeedPPI is the slowest, requiring almost 13 minutes to process a single pair. Such a cost is prohibitive even at this scale and does not encourage its practical adoption. D-SCRIPT's IS and transformer-based models appear relatively affordable on this smaller test set, but the picture changes when scaling to larger interaction spaces: their per-pair cost grows linearly, making them impractical for genome-wide re-ranking. Among them, PubMedBERT is the most demanding due to its cross-encoder stage, while BioBERT and BioMedRoBERTa offer somewhat lighter alternatives. By contrast, functional enrichment and its associated semantic scores are extremely efficient, completing in seconds, and thus remain attractive for large-scale screening.

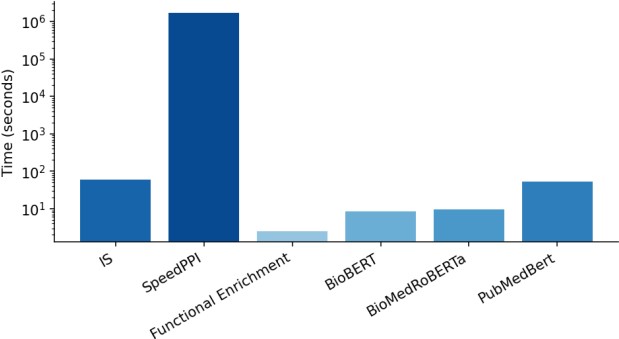

Figure 3: Computational costs of the different re-ranking strategies, reported as total time required to prioritize 2,280 candidate pairs.

## A.3 EVALUATION ON THE PINUI BENCHMARK

Table 4: Retrieval performance at different cutoffs $k$, comparing our interpretability-guided method exploiting active embedding regions against the direct interaction probability ranking from D-SCRIPT on the PiNUI dataset.

| Method | $k$ | Recall↑ | Precision↑ | MAP↑ | nDCG↑ | Success↑ | Pred. Cov.↑ | MRR↑ | Avg. Rank↓ |
|---|---|---|---|---|---|---|---|---|---|
| D-SCRIPT | 5 | 0.000330 | 0.002041 | 0.002041 | 0.003948 | 0.010204 | | | |
| | 10 | 0.000330 | 0.001020 | 0.001020 | 0.003948 | 0.010204 | | | |
| | 50 | 0.003443 | 0.001020 | 0.003443 | 0.011315 | 0.040816 | 0.0080 | 0.0041 | 86.50 |
| | 100 | 0.007493 | 0.001020 | 0.007493 | 0.019548 | 0.091837 | | | |
| | 200 | 0.056388 | 0.000612 | 0.008485 | 0.021796 | 0.112245 | | | |
| | 500 | 0.009335 | 0.000265 | 0.009335 | 0.023035 | 0.122449 | | | |
| Our Approach | 5 | 0.000000 | 0.000000 | 0.000000 | 0.000000 | 0.000000 | | | |
| | 10 | 0.001729 | 0.003061 | 0.003061 | 0.002444 | 0.000000 | | | |
| | 50 | 0.015674 | 0.004286 | 0.015674 | 0.014482 | 0.000000 | 0.3849 | 0.0038 | 924.78 |
| | 100 | 0.028986 | 0.004082 | 0.028986 | 0.022768 | 0.000000 | | | |
| | 200 | 0.056388 | 0.003724 | 0.056388 | 0.038160 | 0.020408 | | | |
| | 500 | 0.133404 | 0.003408 | 0.133404 | 0.074430 | 0.051020 | | | |

To assess whether our interpretability-guided retrieval generalizes beyond STRING-derived data, we further evaluate our method on the PiNUI dataset (Dubourg-Felonneau et al., 2023). PiNUI addresses known biases of classical PPI benchmarks by constructing high-quality positive pairs from IntAct Hermjakob et al. (2004) and nearly uniform negatives that avoid co-expression and localization shortcuts. Unlike STRING, which integrates heterogeneous evidence sources, PiNUI provides a more stringent test of sequence-based and structure-aware generalization. In this setup, D-SCRIPT is applied directly as a probability-based ranking model, whereas our method uses its residue-level activation patterns to guide retrieval, followed by the same evaluation metrics as in the STRING v11→v12 experiments. As shown in Table 4, D-SCRIPT retrieves only a very small fraction of PiNUI positives (Rediscovery Ratio = 0.0080), yielding extremely low Recall@k and MAP@k across all cutoffs. This confirms both the difficulty of the dataset and the limited prospective capability of probability-only predictions. It is important to clarify how Average Rank is computed in the PiNUI evaluation, as this metric is not directly comparable across methods when their rediscovery coverage differs substantially. D-SCRIPT recovers only a very small number of novel interactions, meaning that its Average Rank is computed solely over the proteins for which at least one new partner receives a non-zero score. As a consequence, the reported value reflects a highly restricted subset of the evaluation set and does not represent performance over the full candidate list. By contrast, our interpretability-guided approach retrieves a much larger portion of PiNUI positives (Rediscovery Ratio = 0.3849). Its Average Rank is therefore computed over a significantly larger set of rediscovered instances, which naturally results in a higher numerical value. This difference should not be interpreted as worse ranking quality; rather, it reflects that our method surfaces many more true positives, making the rediscovery set substantially broader. For this reason, Recall@k, MAP@k, nDCG@k, and Rediscovery Ratio provide a more faithful basis for comparing prospective performance across methods.

## B    LLMS STATEMENT

LLMs were employed for grammar checking and text polishing. In addition, domain-specific models such as PubMedBERT, Sentence-BioBERT, and BioMedRoBERT were used as part of the contributions, with details provided in the manuscript. Importantly, LLMs were not used to generate ideas for contributions or to retrieve literature.

