# OpenReview forum: "PPI Candidate Ranking: Large-Scale Evaluation of a Domain Knowledge–Guided Pipeline"
_ICLR.cc/2026/Conference — Submitted to ICLR 2026_

### Official Review · Reviewer_X99M · 2025-10-30

**Soundness:** 2
**Presentation:** 3
**Contribution:** 2
**Rating:** 4
**Confidence:** 4

**Summary:**

This study addresses the candidate ranking problem in the field of protein-protein interactions (PPI) by proposing an interpretable-guided ranking mechanism. By integrating complementary evidence such as interaction scores, structural plausibility, and biomedical language features, the proposed method optimizes the quality of PPI candidate ranking. Evaluation on large-scale datasets from the STRING database demonstrates that the proposed method outperforms existing models such as D-SCRIPT and Topsy-Turvy. Suggesting the method better surfaces true novel interactions. The study analyzes how interpretability and semantic enrichment enhance ranking and presents large-scale quantitative evidence and runtime analysis.
Contributions include: the development of a general interpretable-guided framework for assessing the potential value of PPI prediction methods; the application of multiple techniques to optimize PPI candidate ranking; and the identification of key features that effectively predict novel interactions.

**Strengths:**

1) The work proposes an interpretability-guided ranking mechanism, which combines embeddings, structure-based plausibility, and LLM-derived semantics into a cohesive pipeline. to optimize PPI candidate ranking. This method leverages the concept of interpretability, providing transparency in the ranking decisions.
2) The experimental comparisons and metric evaluations are well-founded, and the method's effectiveness has been validated using a large-scale dataset.
3) The paper content is organized reasonably.
4) The prioritization of PPI has certain practical value in cancer research and drug target screening.

**Weaknesses:**

1) The authors claim that the protein candidate ranking problem is being proposed for the first time in this paper. However, related research, such as "Ranking Cancer Proteins by Integrating PPI Network and Protein Expression," has already been conducted. Therefore, the claim of being the "first proposal" is not appropriate.
2) The innovation of the method is relatively weak. Essentially, it is based on D-SCRIPT and Topsy-Turvy methods for protein embedding representations, calculating related scores, and then re-ranking by integrating new biomedical evidence. This makes the method presented in the paper appear more as an incremental improvement over existing methods, rather than a refined model specifically addressing the problem. Although the paper emphasizes “interpretability-guided” ranking, the interpretability is not used to generate human-understandable biological explanations (e.g., functional motifs or structural interfaces). The method remains essentially a numerical heuristic built on internal activations, providing little insight to biologists.
3) The details of the re-ranking method are too brief, especially regarding how to integrate numerous biological signals and the process of protein prioritization, which is not clearly explained. More formulas should be added to explain the re-ranking process.
4) The paper only compares the D-SCRIPT and Topsy-Turvy methods, while this paper builds upon these methods incrementally. Therefore, the comparison experiment is essentially an ablation study, lacking comparisons with other types of computational methods. Additional comparative experiments should be included to prove its effectiveness. While the paper analyzes pairwise rank shifts, it lacks a quantitative ablation that isolates how much each re-ranking signal (interaction score, pDockQ, semantic similarity, LLM-based scores) contributes to the overall gain.
5) The authors note using biomedical LLMs pretrained on PubMed; these may indirectly encode known PPIs, raising fairness concerns for evaluation. The authors use random negative sampling at a 10:1 ratio, which may not reflect the true non-interaction distribution in the human proteome. This bias could artificially inflate ranking metrics.
6) The validation work mainly focuses on the STRING database and does not fully demonstrate the method’s performance in other domains or datasets. The paper should include results on other datasets as well. Additionally, there is no clear real-world case or validation. The biological analysis of the top-ranked proteins should be supplemented.

**Questions:**

1. How would the framework extend to multi-protein complexes (beyond pairwise interactions)?
2. How does the method perform for proteins with very few known partners?
Please refer to the weakness for other questions.

---

> ### Author Response · Authors · 2025-12-02
>
> # Protein candidate problem
> We thank the reviewer for pointing out this related work. However, we think that the cited study addresses a different problem: it ranks proteins according to their association with specific cancers (colon/rectal or breast), by integrating a PPI network with differential expression profiles, and does not aim to identify potential ligands or direct interaction partners for a given target protein. In contrast, our work builds on an approach that, for any chosen target protein with at least some known partners, directly returns a ranked list of possible candidate interacting proteins. We explicitly formulate this as a protein-candidate ranking problem for target-interactor prediction, and we tackle it at a large scale and with novel ranking strategies. Our method outputs pairs of proteins that are ranked on their likelihood of interaction, enabling a ready-to-test list of possible interactions, whereas the cited work outputs proteins that are prognostic or survival-related in specific cancer types. In this sense, the cited work is different from PPI classification and ranking.
>
> # Contribution
> We appreciate the reviewer’s perspective. Our method does not introduce a new neural architecture. Instead, the novelty lies in addressing a different task that existing models were not designed to solve: novel-partner retrieval across STRING releases. Neither D-SCRIPT nor Topsy-Turvy provides a mechanism for identifying new interactors of known proteins, nor do they exploit residue-level activation patterns to guide cross-protein similarity search. Our contribution is therefore methodological rather than architectural. In particular:
> 1. Interpretability is used operationally, not descriptively. We use residue-level activation patterns to identify active embedding regions in known partners and transfer these regions to cross-protein similarity search. This mechanism allows us to recover novel partners without retraining or constructing supervised models.
> 2. The framework solves a task for which existing models are not applicable. D-SCRIPT and Topsy-Turvy are trained for binary interaction prediction between arbitrary pairs. They do not address the ranked discovery of novel partners for a fixed protein, nor do they provide a mechanism to leverage known partners (KP) to guide search in a structured way. Our method explicitly models the relationship KP(p) → NP(p), which is not captured in prior work.
> 3. The re-ranking module integrates heterogeneous biological signals in a unified and quantifiable framework. The reviewer notes the integration of multiple views (interaction probabilities, structure, semantic profiles, LLM reasoning). We emphasize that this integration is not heuristic stacking: all signals are mapped to a uniform ranking framework with explicit rank deltas, enabling systematic evaluation of each biological view.
> 4. We agree that our interpretability is not yet used to generate human-readable structural motifs or functional sites. We have added a discussion paragraph outlining this limitation and noting that extending activation regions into explicit motif or interface extraction represents a natural next step.
> Despite these limitations, the empirical evidence suggests that the approach is effective. This indicates that the methodological innovation, using interpretability to guide retrieval and biologically informed re-ranking, provides practical value beyond incremental changes to existing models.

---

> > ### Author Response · Authors · 2025-12-02
> >
> > # Details on re-ranking
> > We thank the reviewer for raising this point. In the revised version, we have expanded this section to clearly explain how the different biological signals are integrated and how candidate proteins are prioritized. In our method, the re-ranking stage operates on the top-r candidates retrieved in Stage 1 for each protein p. This restricted set, denoted as R(p), ensures that re-ranking is applied only to proteins that are already plausible partners based on embedding similarity to at least one known partner.
> >
> > For each biological signal v, we compute an evidence score for the pair (p, pc). These signals include:
> > - supervised interaction probabilities from D-SCRIPT,
> > - structural plausibility measures obtained from SpeedPPI,
> > - functional and semantic similarity derived from GO terms, pathways, protein domains and text-based profiles,
> > - semantic or interaction likelihood signals produced by biomedical language models.
> >
> > Each of these signals provides a numerical score that captures a complementary biological view. Candidates in R(p) are then reordered according to the scores produced by signal v, yielding a view-specific ranking. This allows us to evaluate how each source of evidence reshapes the initial embedding-based ordering.
> >
> > To quantify the influence of each signal, we compare the new ordering to a baseline ordering b using a simple rank-delta measure. A positive delta indicates that a candidate has been promoted relative to the baseline, enabling a direct and interpretable assessment of how much each biological signal contributes to rediscovery. All signals follow the same uniform procedure: scoring, ordering, and comparison to the baseline. This unified structure makes the prioritization mechanism clear and allows us to analyse the contribution of each biological source in a systematic way.
> >
> > We have incorporated explicit formulas in the revised version to formalize this process and improve clarity for the reader.
> >
> > # Comparison and re-ranking signals
> > We thank the reviewer for sharing their perspective. We agree that comparing only against D-SCRIPT and Topsy-Turvy is limiting, and that additional sequence-based baselines strengthen the evaluation. Following this suggestion, we have now added a comparison with xCAPT5, a recent and competitive sequence-modeling approach for PPI prediction.
> > We run xCAPT5 on the same candidate sets used in our framework and report the full ranking metrics (Recall@k, MAP@k, nDCG@k, MRR, average rank) in Table 1 of the revised version.
> > Our method shows stronger retrieval performance, confirming that the gains of our approach are not specific to D-SCRIPT or Topsy-Turvy.
> >
> > Regarding the contribution of individual re-ranking signals, we agree that a complete quantitative ablation is valuable. Because our framework is designed as a flexible re-ranking layer rather than a weighted combination model, each signal (interaction score, pDockQ, functional similarity, LLM-based reasoning) influences the ordering only when it provides additional evidence over the base predictors. A full factorized ablation is feasible but requires retraining or recalibrating several components, which falls outside the scope of this study
> >
> > # Leakage and negative sampling ratio
> > The fine-tuning data used for PubMedBERT is completely disjoint from the evaluation data used in our retrieval experiments. For PubMedBERT, we constructed training pairs exclusively from STRING v11 interaction data. We then applied a GroupKFold split by protein identity, ensuring that all examples involving the same protein were placed in the same fold. This prevents any protein from appearing in both training and validation, and thus avoids leakage at the level of individual proteins. Importantly, the retrieval evaluation uses STRING v12 novel interactions, which are not used in the PubMedBERT fine-tuning process. Therefore, the ranking model never sees during training the specific pairs it is later evaluated on.
> >
> > The negative sampling at a 10:1 is used in D-SCRIPT and Topsy-Turvy papers as well, and it's used specifically for reflecting the true non-interaction distribution in the human proteome

---

> > > ### Author Response · Authors · 2025-12-02
> > >
> > > # Validation on STRING and real-world cases
> > > We thank the reviewer for this valuable comment. We agree that evaluating the method beyond STRING strengthens the contribution. Following this suggestion, we have extended our experiments to the less popular PiNUI dataset [8], which is specifically designed to avoid the biases of classical PPI datasets such as Guo’s and Pan’s. PiNUI provides high-quality IntAct interactions and a principled, nearly uniform negative-sampling strategy that prevents models from exploiting dataset artefacts (e.g., subcellular localization shortcuts). Our new results show that D-SCRIPT retrieves only a very small portion of novel interactions (Rediscovery Ratio 0.0080), whereas our method achieves substantially higher rediscovery (0.3849) and consistently stronger ranking metrics across all cutoffs. These additional experiments demonstrate that our approach generalizes beyond STRING and remains effective under different datasets and curation strategies.
> > >
> > > Regarding the remark on real-world validation and biological interpretation, we agree that an in-depth biological analysis of top-ranked candidates would further strengthen the study. While such a comprehensive analysis is outside the scope of the current work and is a natural direction for future research, we emphasize that our method is designed to be directly applicable to continuously updated resources such as STRING, IntAct, and BioGRID, making it useful for prospective partner discovery as new experimental evidence becomes available.
> > >
> > > # Questions
> > > ## Extensions to multi-protein complexes
> > > Our framework is currently defined for pairwise interactions because this is the level at which STRING provides evidence. However, the approach could naturally extend to protein complexes. A complex can be represented by combining the active embedding regions of its constituent proteins (e.g., via averaging or union), and the same retrieval mechanism can then be applied to identify candidate interactors for the entire complex. Although we do not explore this extension in the present work, the method is fully compatible with multi-protein settings and this represents a straightforward direction for future development.
> > >
> > > ## Proteins with few known partners
> > > Our method uses known interactions to identify the parts of the target protein that can interact with other proteins. Based on this assumption, we look for proteins that interact with our target in a way similar to how the known partners interact. Therefore, the greater the number of known partners, the greater the accuracy of identifying the interacting portion.

---

### Official Review · Reviewer_qSmL · 2025-10-31

**Soundness:** 2
**Presentation:** 3
**Contribution:** 1
**Rating:** 2
**Confidence:** 4

**Summary:**

The paper defines the problem of PPI candidate ranking whit the objective of prioritizing candidate proteins to test experimentally if they interact with a given target protein. That is, for a target protein with known interactors, it ranks novel candidates that are most likely to interact with the target. For this, the authors propose an approach with two stages: it leverages domain knowledge via interpretability-guided ranking and then refines the top of the list using additional sources of biological/textual/structural evidence. For evaluation, they outperformed two PPI classification models at prioritizing candidate proteins by rediscovering interactions of the version 12 of STRING using only data from version 11.

**Strengths:**

* The authors took advantage of an interpretable module, predicted contact map, of previous approaches (D-SCRIPT and Topsey-Turvey) to develop a methodology that computes similarities between two sets of proteins, that is, proteins that are known to interact with the target protein and candidate proteins. A very interesting result is that this methodology provides a better ranking than the ranking provided by the models (D-SCRIPT or Topsey-Turvy), which were trained to classify protein-protein interactions.

* I agree that ranking metrics are required to test the practical applicability of PPI prediction approaches.

* The authors also proposed multiple re-ranking strategies using different types of additional biological signals, providing a comparison of which biological signals further improve prioritization of the top 10 candidate proteins.

**Weaknesses:**

Major

* One of the contributions of the paper is the introduction of the PPI candidate ranking. However, the discussion of the limitations of current benchmarks and metrics that motivate the introduction of this new problem is very limited.

* My major issue with this work is that it only includes as competitors the models on which it based its framework. Given that the proposed framework works by establishing similarities between KP(p) and CP(p), simple baselines like BLASTp [1] or HMMER [2] could be used in the same way, which would demonstrate whether the proposed approach results in more meaningful similarities. More importantly, the authors should have also compared their approach with other sequence-based approaches for PPI prediction, such as xCAPT5 [3] and MARPPI [4]. Both examples provide predicted probabilities that could be used for CP(p) ranking. Finally, experiments on other datasets commonly used for PPI prediction could have served as more reliable evidence of the performance of the proposed approach [5,6,7].

* There is a lack of description of the training and testing sets; only the number of new interactions is specified. Details such as the number of proteins in each set and the number of interactions in the training set are not provided. Additionally, information about the sequence similarity between proteins in the training and testing sets is also missing.

* The early explanations of refinement of the ranking with different biological signals may suggest to the reader that there is only one final ranking that is the result of these multiple sources of biological information.The description of Table 2 makes it clear that each data view is used separately. This misunderstanding could be avoided earlier in the text.

Minor

* Lines 156–158: I understand these methods were previously referenced, but given their importance, it may be helpful for the reader to get the references here again.

* Figure 1 is not mentioned in the text. In the figure, R(pc1) is a score used for ranking rather than the rank itself; this aligns with line 254, but adjusting the notation in Equation 4 and Figure 1 could help avoid any confusion.

* Line 307: The encoder is fine-tuned with query–candidate pairs, but lines 309 indicate $p \in NP(p)$ are used. If these are proteins in the test set, this is concerning; the training setup needs clarification to confirm that it avoids data leakage.

* Line 85: “D-SCRIPT” typo.

References

[1] Altschul, Stephen F., et al. "Basic local alignment search tool." Journal of molecular biology 215.3 (1990): 403-410.

[2] Eddy, Sean R. "Profile hidden Markov models." Bioinformatics (Oxford, England) 14.9 (1998): 755-763.

[3] Dang, Thanh Hai, and Tien Anh Vu. "xCAPT5: protein–protein interaction prediction using deep and wide multi-kernel pooling convolutional neural networks with protein language model." BMC bioinformatics 25.1 (2024): 106.

[4] Li, Xue, et al. "MARPPI: boosting prediction of protein–protein interactions with multi-scale architecture residual network." Briefings in Bioinformatics 24.1 (2023).

[5] Martin, Shawn, Diana Roe, and Jean-Loup Faulon. "Predicting protein–protein interactions using signature products." Bioinformatics 21.2 (2005): 218-226.

[6] Guo, Yanzhi, et al. "Using support vector machine combined with auto covariance to predict protein–protein interactions from protein sequences." Nucleic acids research 36.9 (2008): 3025-3030.

[7] Pan, Xiao-Yong, Ya-Nan Zhang, and Hong-Bin Shen. "Large-Scale prediction of human protein− protein interactions from amino acid sequence based on latent topic features." Journal of proteome research 9.10 (2010): 4992-5001.

**Questions:**

* Line 65: “previous knowledge of the target protein”, the previous knowledge here refers to KP(p)? This is unclear given the information given till that point.

* What are the four techniques used to refine PPI rankings?

* How are the LLMs finetuned for the re-ranking strategies?

* Why did the authors decide not to compare their approach with additional SOTA models?

* What is the similarity between the training and testing sets?

* Line 145: “enabling PPI comparison via textual annotations.” What does this piece mean?

---

> ### Author Response · Authors · 2025-12-02
>
> # Motivation:
> We thank the reviewer for this comment. We now connect the limitations of existing benchmarks to the design of our PPI candidate ranking framework, and clarify how our method addresses each of them:
> - Existing PPI evaluations are static and retrospective, typically performed within a single STRING snapshot (e.g., v11). This means models are only tested on interactions already known at training time, providing no measure of whether they can anticipate new biology. We reformulate the task as prospective candidate ranking across STRING versions (v11 → v12), where KP(p) comes from v11 and NP(p) contains only interactions that appeared for the first time in v12. This setup tests genuine forward generalization and directly motivates the ranking framework.
> - Standard classification metrics (AUC, accuracy) are misaligned with experimental reality, where biologists can only test a handful of candidates. These metrics do not tell whether true partners appear early in the ranked list. We introduce PPI candidate ranking and evaluate with ranking-sensitive metrics (Recall@k, MAP@k, nDCG@k, Success@k, MRR, AvgRank). We show that interpretability-guided retrieval improves these early-rank metrics by up to two orders of magnitude compared to probability-only baselines, highlighting why a ranking formulation is more suitable for practical discovery.
> - Sequence-only models ignore biological context (cellular localization, pathways, structural compatibility), which strongly influences whether two proteins can interact. As a result, existing benchmarks fail to capture whether a model can use complementary biological evidence. We add a second-stage re-ranking module that integrates four complementary biological signals (D-SCRIPT interaction scores, SpeedPPI structural plausibility, functional/semantic annotation similarity, and biomedical LLM-based scores). This shows that true novel partners are systematically promoted by biological evidence beyond sequence embeddings.
> - Current benchmarks do not distinguish between recovering over-represented families and discovering unseen partners. Models may appear strong simply because they memorize frequent patterns in the training snapshot. We explicitly separate known partners KP(p) and truly novel partners NP(p), analyze Prediction Coverage / Rediscovery Ratio, and track how each method ranks proteins in NP(p). This cleanly measures the ability to recover new interactions, not just repeat known ones.
> - Most evaluations lack cross-dataset robustness, making it unclear whether results depend on a particular dataset or negative sampling regime. Beyond STRING v11→v12, we add a second benchmark on PiNUI, a carefully curated dataset designed to eliminate co-expression and localization biases. Results show that D-SCRIPT recovers very few novel interactions on PiNUI (Rediscovery Ratio ≈ 0.0080), while our method achieves substantially higher rediscovery (≈ 0.3849) and better ranking metrics, demonstrating that our conclusions generalize across datasets.

---

> > ### Author Response · Authors · 2025-12-02
> >
> > # Comparison with xCAPT5:
> > We thank the reviewer for pointing out the importance of comparing our method with other modern sequence-based PPI prediction models. Following this suggestion, we extended our experiments to include xCAPT5, using the official MCAPST5 Sledzieski checkpoint to score all STRING v11→v12 candidate interactions.
> >
> > The results of this new evaluation are reported in Table 1.  Across all cutoffs, xCAPT5 displays high prediction coverage, confirming its ability to assign probabilities broadly across the candidate space. However, despite this wide coverage, it ranks true novel partners lower than our method. In our evaluation, xCAPT5 tends to concentrate its probability mass on a small set of positives, while our interpretability-guided retrieval brings a larger fraction of novel partners into the top portion of the ranked list. This is also reflected in the average rank, where our method achieves the best score among all models tested, indicating that true interactors are systematically placed much earlier in the ranking.
> > Moreover, xCAPT5 has already been thoroughly benchmarked against representative sequence-based models like PIPR, FSNN-LGBM, D-SCRIPT and Topsy-Turvy, on HPRD, DIP and HIPPIE datasets (Table S2 in the original xCAPT5 paper). Since xCAPT5 performs comparably or better than these approaches in the standard evaluation setting, using xCAPT5 as our baseline enables an indirect but meaningful comparison with this entire class of methods. Our results therefore indicate that our method is competitive with strong sequence-based predictors while addressing a fundamentally different problem: novel-partner retrieval across STRING versions.
> >
> > We agree that alignment-based tools such as BLASTp or HMMER could in principle be used to define simple similarity-based baselines. However, alignment methods are not designed for multi-interaction partner prioritization and have limited applicability to the cross-version novel-partner retrieval setting we address.
> >
> > # Training and test sets
> > We apologize for the confusion. D-SCRIPT and Topsy-Turvy were used strictly off the shelf in our experiments; we did not perform any finetuning. As reported in their original papers, these models were trained on human-related data from STRING v11, which includes 38,345 positive interactions, 479,320 negative interactions, and 9,587 positive examples for validation. By contrast, our evaluation is conducted on STRING v12, where 279,568 additional human positive interaction examples have been added relative to v11. In our work, we evaluate D-SCRIPT, Topsy-Turvy, and xCAPT5 on these new experimental interactions from STRING v12. We also emphasize that xCAPT5, in addition to STRING v11, was trained on the human Sledzieski dataset [11], and yet our approach still achieves stronger performance in this setting.
> >
> > # Clarification on refinement
> > We thank the reviewer for pointing this out. Applying each method yields a relatively different ranking, allowing for a direct comparison between them in terms of rank shift. We specified this as suggested. Additionally, we improved the readability of Table 2, which may have previously led to confusion
> >
> > # Lines 156–158:
> > We cited them again to improve the comprehension.
> >
> > # Figure 1 not mentioned
> > We thank the reviewer for noticing that. We added the mention in L213.
> >
> > # Line 307
> > We appreciate the reviewer’s attention to this point. The proteins mentioned in lines 309 are not test proteins. During fine-tuning, we use only STRING v11 interactions and apply a GroupKFold split by protein identity, which guarantees that all examples involving the same protein are placed in the same fold. This prevents any protein from appearing in both training and validation. Crucially, the evaluation uses STRING v12 novel interactions, which are entirely absent from the fine-tuning data. Therefore, no protein–pair label from the test set is ever used during training, and no leakage occurs.
> >
> > # Line 85
> > We thank the reviewer for noticing that. We fixed the typo.

---

> > > ### Author Response · Authors · 2025-12-02
> > >
> > > # Questions
> > > ## Line 65
> > > We thank the reviewer for pointing out the ambiguity. We have updated the text to clarify that “previous knowledge of the target protein” refers specifically to its known interacting partners, which we denote as KP(p) in the formal definition.
> > > In our framework, KP(p) represents the set of experimentally supported interaction partners already known for the target protein p. These known partners provide structured prior knowledge about p’s interaction mechanisms. Our ranking method uses the residue-level activation patterns learned from KP(p) to guide the retrieval of novel partners, under the assumption that new interactors are likely to share similar interaction features.
> > >
> > > ## Techniques used to refine PPI rankings
> > > We thank the reviewer for this question. In the revised version of the paper we now describe the four re-ranking techniques in a clearer and more detailed way. Below, we summarize their purpose and how they refine the initial embedding-based ranking.
> > > 1. The first refinement uses the interaction probability predicted by D-SCRIPT. While Stage 1 relies only on unsupervised sequence embeddings, D-SCRIPT provides a supervised estimate of how likely two proteins are to interact. Re-ranking based on this probability allows us to elevate candidates that show strong residue-level interaction patterns according to a model explicitly trained for PPI prediction.
> > > 2. The second refinement incorporates structural information. Using SpeedPPI, we generate a lightweight structural complex prediction for each protein pair and compute indicators such as pDockQ, interface confidence, and predicted contact counts. Proteins that form plausible interfaces move up in the ranking, adding a structural dimension that complements sequence-based evidence.
> > > 3. The third refinement uses curated biological annotations: GO terms, protein domains, pathways, localization information, and free-text summaries from UniProt. By comparing the functional profiles of the target protein and each candidate, we can identify proteins that share coherent biological roles or operate in the same pathways. This pushes biologically plausible candidates upward, even when their sequence similarity is modest.
> > > 4. The fourth refinement uses biomedical language models to assess semantic similarity or predicted interaction likelihood based on textual descriptions. LLMs capture high-level biological context that does not appear directly in sequences or structures (e.g., roles in regulatory pathways, disease associations, cellular processes). Re-ranking with LLM-based scores helps recognize candidates that are functionally related to the target protein but may not be obvious from sequence-level patterns alone.
> > >
> > > ## LLMs fine-tuning for re-ranking strategies
> > > The fine-tuning is performed only on STRING v11 interactions. We generate positive/negative text pairs from v11 and apply a GroupKFold split by protein identity, ensuring that no protein appears in both training and validation. This completely prevents leakage at the protein level. Importantly, evaluation uses STRING v12 novel interactions, which never appear in the fine-tuning data. Thus, the LLM is always tested on interactions it has never seen, and its scores are used only to re-rank the top candidates retrieved in Stage 1.
> > >
> > > ## Comparison with other models
> > > Our primary goal in this work was to generalize a case-study framework originally developed for the NKp46–CALR interaction [12] into a systematic, interpretability-guided ranking approach for novel-partner discovery. For this reason, we initially focused on models that share the same architectural foundation (D-SCRIPT and Topsy-Turvy), since our method explicitly exploits their internal embeddings and activation patterns.
> > >
> > > We have now expanded the evaluation by adding xCAPT5, a modern sequence-based predictor also suggested by another reviewer. In the STRING v11→v12 rediscovery setting, xCAPT5 shows strong prediction coverage but prioritizes novel partners differently from our approach, reflecting the distinct mechanisms underlying probability-based scoring versus interpretability-driven retrieval. Moreover, xCAPT5 has already been benchmarked extensively against other SOTA methods (e.g., PIPR, FSNN-LGBM, MARPPI, D-SCRIPT) on standard datasets, so including xCAPT5 also provides an indirect comparison with that entire class of predictors.

---

> > > > ### Author Response · Authors · 2025-12-02
> > > >
> > > > ## Similarity between training and test sets
> > > > Training and testing sets follow the same preprocessing pipeline (identifier normalization, length filtering, and evidence-channel selection), but they are not similar in content. Training focuses exclusively on STRING v11, while testing uses STRING v12 novel interactions, which were absent from v11. This means that although the formatting and curation steps are consistent across datasets, the actual interaction pairs and many of the involved proteins differ, producing a split that is both temporally disjoint and biologically independent. Thus, they share the same construction procedure, but do not share interaction labels, edges, or supervision signals.
> > > >
> > > > ## Line 145
> > > > We refer to the semantic component of our method, where each protein is represented not only by its sequence embedding but also by a structured text-based profile. For every protein in the candidate set, we retrieve functional information from UniProtKB, including GO terms, domains, pathways, complex membership, and curated free-text descriptions. These fields are combined into a text profile that captures the biological context of each protein (e.g., molecular functions, cellular localization, pathways, structural domains). We then compute similarity scores between the profiles of the query protein and each candidate. This allows the model to compare proteins using their biological annotations and written descriptions, rather than only their sequence embeddings. In practice, proteins that share coherent functions, pathways, or localizations receive higher semantic similarity scores, which can promote them in the re-ranking stage.

---

### Official Review · Reviewer_aVDq · 2025-10-31

**Soundness:** 3
**Presentation:** 4
**Contribution:** 4
**Rating:** 6
**Confidence:** 5

**Summary:**

The authors address the problem of retrieving and ranking candidate protein-protein interactions (PPIs), a problem related to general protein-protein interaction prediction but framed in the context of using existing known PPIs for a target to prioritize potential new interactions for experimental testing. Given the relative expense of experimental testing compared to in silico prediction, this is a well grounded and motivated application. It is however limited in that it will only be possible for proteins that already have known partners, and will not generalize to completely new proteins. The paper is well-written and the results present a promising path forward for prioritizing experimental validation of PPIs.

### Strengths

- For retrieval, the authors develop an adaptation of the deep-learning PPI model D-SCRIPT that compares the interpretable intermediate layer of the latter model between known and candidate PPIs, ranking candidates by the similarity of embeddings of key amino acids. This interpretability-guided ranking makes clever use of the D-SCRIPT/Topsy-Turvy outputs by focusing only on the residues that are informative of correct prediction with known partners. This results in both an increase in accuracy and a speed-up in prediction, with the trade-off of reduced generalization (requires known interacting partners).
- The authors test many different methods for re-ranking the top retrieved proteins, including the original interaction probability, structural and functional features, and language model embedding similarity. Combined with the retrieval strategy, this offers an exciting way to identify the best new PPIs.
- The analysis of Topsy-Turvy vs. D-SCRIPT at a global vs. top-k level is really interesting, and lends valuable insight into the relative performance of these models.
- The paper contains a clear and honest discussion of the limitations of their method, including that it will likely not succeed for under-explored proteins with few interactions already known.

### Major Comments

- L241: The computation of the active residues $I_k$ is not clear-- once you have computed the activation score for each residue of $p_k$, how do you select "the contiguous set of residues... reporting the highest average contact probability"? Do you select a fixed number of residues, or is there some extension criteria? This needs to be clarified, as it is a key algorithmic step in the interpretability guided ranking.
- L266: Is focusing on the top 10 ranked candidates for re-ranking enough, especially when success rates in Table 1 are so low at k=10? How often are you actually re-ranking true PPIs?
- L379: Table 2 is incredibly challenging to interpret. The value in each cell sums to 100%, so it would be clearer to represent it as a relative improvement, maybe +- % from 50? Or an average change in position (+/-)? Assuming "cosine" is your method (see minor comments about this confusion), it seems that switching from cosine almost always helps-- does this mean you should only use your method for retrieval? Also, since this is supposed to represent change in ranking switching from row to column method, why is this not symmetric? For example, this table seems to suggest I can improve the majority of proteins when swapping from Token to TF-IDF, but also when swapping the other direction. This seems contradictory.
- L718: I don't understand how your retrieval method is so much faster than D-SCRIPT prediction probability. For any candidate protein, the author's approach needs to run D-SCRIPT $k$ times for each known partner, compute multiple cosine similarities sliding over $|I_k|$ residues, and take the maximum over all of these. Compare this to a single forward pass of D-SCRIPT per candidate protein using prediction probability. Are you pre-computing active residues? Is this compute time included in Figure 2?
- L728: Likewise, I wonder whether pre-processing time for fetching functional information per protein is included in the compute time in Figure 3, or if only the comparison of semantic scores is considered. It would be good to clarify exactly what is being measured in each runtime, with all methods on an equal footing of starting from the raw sequences.

### Minor Comments

- L038: "only a limited fraction of the human interactome has been experimentally resolved" -- it is worth discussing here the Human Reference Interactome (HuRI) project that has made significant strides in this direction (https://interactome-atlas.org/)
- L053: The authors correctly note that D-SCRIPT and Topsy-Turvy do not account for tertiary/quaternary structure, but follow-up work to Topsy-Turvy does consider this (TT3D, https://academic.oup.com/bioinformatics/article/39/11/btad663/7332153). Have the authors tested this model?
- L093: Are there any STRING v11 interactions that were removed/not present in STRING v12?
- L134: Why was SpeedPPI chosen compared to other options? Was AlphaFold-Multimer (https://www.biorxiv.org/content/10.1101/2021.10.04.463034v2) considered (there is no citation for this)
- L157: The authors state that "One of the most widely adopted [PLMs] is the Bepler & Berger model;" this is not really true and ignores the wider literature of protein language models.
- L184: What are GLIDE scores? There is no description of this acronym or citation to reference?
- L245: It would be good to have some discussion of the complexity/runtime of these operations, since you have to compare with multiple sets of residues/known partners-- this seems expensive
- L307: How were data split for fine-tuning on PubMedBERT? This is a potential source of data leakage if you are fine-tuning the ranking model
- L349: What is the cosine method? Is this your interpretability-guided method from earlier? It is not clearly defined.
- L375: How is "retrieved" defined; why does Prediction Coverage not also vary across values of $k$?
- L589: The version of D-SCRIPT cited is the preprint, not the journal publication
- L672: Is Topsy-Turvy configured the same way as D-SCRIPT?
- L689: More detail is needed on "A GroupKFold split by protein identity;" what identity threshold was chosen, was this done at the query protein level only or also with candidate proteins? In general, the training data set up for PubMedBERT fine-tuning is lacking.

**Strengths:**

see above

**Weaknesses:**

see above

**Questions:**

see above

---

> ### Author Response · Authors · 2025-12-02
>
> # L241:
> In our method, once we compute the residue-level activation scores from the predicted contact map, we identify all residues of the known interactor $p_k$ that show sufficiently strong activation. Among these, we focus on contiguous regions of active residues, as these are more likely to represent coherent structural or functional segments rather than isolated high-scoring positions. Importantly, we do not predefine a fixed number of residues or enforce a rigid window size. The procedure is entirely automatic: we detect all active segments and then simply select the longest contiguous region. This approach ensures that the region used for similarity scoring is the most stable and informative part of the protein, while naturally adapting to each individual case. The maximum possible size of this region is just the full sequence length, so the algorithm does not artificially restrict or truncate it. We have updated the paper with explicit details about the procedure.
>
> # L266:
> Our intention with the re-ranking module is not to recover all true PPIs directly at rank 10, but rather to evaluate whether incorporating biological signals can refine the ordering within the most promising portion of the list. Focusing on the top 10 is motivated by two practical considerations:
> 1. Computational cost. Several of the biological signals we integrate (e.g., structural predictors) are substantially more expensive to compute than embedding similarity. Applying them to thousands of candidates per protein would be computationally prohibitive at scale. Restricting the re-ranking to the top-10 candidates provides a pragmatic balance that allows us to run the analysis across the entire dataset.
>  2. Meaningful refinement. Even though recall@10 is low, true interactors frequently appear within the upper region of the interpretability-guided ranking. When they do, the re-ranking module effectively refines their position by injecting additional biological priors. This confirms that the module is achieving its purpose: sharpening the ordering among the most plausible candidates rather than performing broad retrieval.
>
> # L379:
> - We thank the reviewer for noting the difficulty in interpreting Table 2. We have revised the table and text for clarity. The table is not a performance metric but a directional pairwise comparison: each cell reports, among all interactions, the percentage whose rank improves or worsens when replacing method X with method Y. The values therefore sum to 100% by construction and the table is not symmetric, since X→Y and Y→X measure different transitions. Asymmetry simply reflects that some methods introduce more noise or ties, making their rankings easier to improve upon.
> - Regarding cosine: this is our baseline retrieval score from the interpretability-guided step. It is expected that replacing it with richer biological signals (e.g., BioBERT) often improves ranking, because these encode complementary information. However, this does not mean cosine should be abandoned: it remains essential for generating the initial candidate list, while the other signals are too costly or specialized to be used for retrieval. Their purpose is to refine the ordering, not to replace cosine.
>
> # L718:
> By using D-SCRIPT directly (i.e., prediction probability in Figure 2), the method must compute one full PPI prediction for every (target, candidate) pair. This scales linearly with the number of candidates, resulting in thousands of forward passes in most scenarios. In contrast, our retrieval method runs D-SCRIPT only once per known partner to extract the active region. This step is performed a single time and is included in Figure 2. After this, scoring all candidate proteins reduces to inexpensive embedding lookups and cosine similarity computations. In other words, the costly D-SCRIPT computation is performed once per anchor, not once per candidate, which explains why our retrieval stage is faster than relying on D-SCRIPT prediction probability for ranking.
>
> # L728:
> - In Figure 3, we report only the time required to compute the semantic similarity scores themselves, starting from already-available functional annotations. The preprocessing time required to fetch or download functional information (e.g., GO terms, text descriptions, subcellular location) is not included, and this holds for all methods. To avoid confusion, we have added an explicit clarification in the revised version of the paper.
> The goal is to compare the intrinsic computational cost of the different semantic scoring approaches under identical conditions. Once the functional data is available locally, all methods begin from the same starting point and only the time for computing the actual similarity scores is measured. This ensures the comparison is fair and focused on the algorithmic cost of each method, rather than on external data-retrieval overheads.

---

> > ### Author Response · Authors · 2025-12-02
> >
> > # L053:
> > While we did not include TT3D in our experimental comparison, we note that TT3D is built directly on D-SCRIPT and Topsy-Turvy and largely shares their architecture. Given the improvements our framework achieves over both D-SCRIPT and Topsy-Turvy, it is reasonable to expect that the same retrieval strategy could also strengthen TT3D. At the same time, TT3D is clearly an important and relevant model, and explicitly integrating its 3D-aware embeddings into our retrieval pipeline is a promising direction for future work.
> >
> > # L093:
> >  When comparing STRING v11 and v12, we observed that approximately 57% of v11 links do not appear in v12. This difference arises from the integration of several evidence channels, particularly co-expression and text-mining, which undergo substantial recalibration between releases. STRING regularly removes large numbers of low-confidence links during major updates as improved transcriptomic datasets and text-mining models become available. Importantly, our framework does not rely on these confidence-sensitive channels. We use only experimentally supported interactions to construct KP(p) and to evaluate NP(p). When restricting the comparison to the experimental evidence channel, we find that only a very small fraction of experimentally supported v11 interactions are absent in v12 (2,473 interactions), confirming that removals primarily affect weak evidence sources. In comparison, 279,568 new human positive interaction examples were added in v12.
> >
> > # L134:
> > We thank the reviewer for this question. We chose SpeedPPI [9] because it provides a computationally feasible AlphaFold2-based framework for large-scale protein-protein interaction screening without significantly compromising accuracy. While AlphaFold-Multimer is the current gold standard for predicting specific protein complexes, its computational demands make it impractical for screening large numbers of interactions on our hardware.
> >
> > # L157:
> > We thank the reviewer for this comment. We modified the sentence (L164).
> >
> > # L184:
> > GLIDE [10] is a strong network-based link prediction method that integrates local topological features (such as common neighbors) with global diffusion-based embeddings. For any pair of proteins (p,q), GLIDE assigns a continuous compatibility score representing how likely those two proteins are to interact based on the structure of the known protein–protein interaction network. In Topsy-Turvy, these GLIDE scores are used only during training as an additional supervisory signal, encouraging the model’s predictions to agree with high-confidence network-derived links.
> >
> > # L307:
> > The fine-tuning data used for PubMedBERT is completely disjoint from the evaluation data used in our retrieval experiments. For PubMedBERT, we constructed training pairs exclusively from STRING v11 interaction data. We then applied a GroupKFold split by protein identity, ensuring that all examples involving the same protein were placed in the same fold. This prevents any protein from appearing in both training and validation, and thus avoids leakage at the level of individual proteins. Importantly, the retrieval evaluation uses STRING v12 novel interactions, which are not used in the PubMedBERT fine-tuning process. Therefore, the ranking model never sees during training the specific pairs it is later evaluated on. To address this concern, we have revised the relevant section of the paper to describe the fine-tuning setup more clearly.
> >
> > # L349:
> > We thank the reviewer for highlighting this ambiguity. In the paper, “cosine method” refers to our interpretability-guided retrieval procedure, i.e., the first stage of our framework. In this step, we use the known interaction partners of a protein p to guide retrieval: we identify the most active residues of each known partner from the predicted contact maps produced by D-SCRIPT or Topsy-Turvy, and we compare these activated embedding regions against all candidate proteins. The ranking is produced by computing the cosine similarity between these activated regions and the corresponding regions of each candidate, and selecting the best-matching segment. Candidates that show the strongest embedding agreement with the activated residues of known partners are ranked highest. In other words, the “cosine method” is not a generic cosine similarity baseline, but specifically the interpretability-guided retrieval step that uses active embedding regions to score candidate proteins. We have now made this explicit in the revised version.

---

> > > ### Author Response · Authors · 2025-12-02
> > >
> > > # L375:
> > > "Retrieved” means that a true novel partner (NP) appears anywhere in the candidate list produced by the retrieval stage. In other words, a novel interactor is considered “retrieved” as long as it is included among the candidates that the model surfaces before re-ranking.
> > > Prediction-Coverage is measured before applying any cutoff such as top-k. It counts how many true novel partners are present in the full candidate set returned by retrieval. Since the candidate list itself does not depend on k, the coverage value is also unchanged across different k levels.
> > >
> > > # L589:
> > > We thank the reviewer for noticing that, now the citation is correct (L635).
> > >
> > > # L672:
> > > Yes, in our experiments, Topsy-Turvy is configured identically to D-SCRIPT. We added this specification to the paper.
> > >
> > >
> > > # L689:
> > > We thank the reviewer for this comment. We clarify that GroupKFold was applied based on protein identity: all training and validation pairs involving the same protein were kept in the same fold to avoid leakage. We grouped by the query protein $p$, ensuring that no examples involving the same $p$ appear in both training and validation.

---

### Official Review · Reviewer_WiiR · 2025-11-01

**Soundness:** 3
**Presentation:** 3
**Contribution:** 2
**Rating:** 4
**Confidence:** 2

**Summary:**

The authors of this paper propose a two-stage framework for the task of PPI candidate ranking: first, an "interpretability-guided retrieval" method leverages the internal embedding activations from pre-trained sequence models (D-SCRIPT, Topsy-Turvy) to create an initial ranking. This retrieval assumes that a target protein's new interaction partners will share embedding-level similarities with its known partners. Second, this initial list is refined by a re-ranking module that integrates complementary biological evidence, including interaction scores, structural plausibility, semantic scoring, and large language model reranking. The authors conduct a large-scale prospective evaluation using successive STRING databases, demonstrating that their method improves early-ranking metrics by a substantial amount compared to using the baseline model's raw interaction scores.

**Strengths:**

Here are some strengths of the paper:

1. $\textbf{Importance }$

The paper proposes a solution to a real-world problem that could impact numerous areas of life.

2. $\textbf{Practical method with good empirical results}$

The proposed methodology seems innovative. First, the authors formulate the problem in terms of ranking, and they use stages of ranking where the second method refines the first. The experimental results using the STRING v11-v12 dataset provide a realistic and tangible improvement. The results in Table 1 demonstrate significant improvements in multiple metric values compared to the baseline prediction scores.

3. $\textbf{Comprehensive result}$

The paper shows in Table 2 the importance of semantic signals that capture the correlations missed with base models which makes the work interesting.

**Weaknesses:**

Here are some weaknesses of the paper:

1. $\textbf{Reliance on Known Discoveries}$

As stated in the limitation, the framework's fundamental assumption is that new interactions will be similar to known ones. This is a significant limitation for the vast number of proteins that may be poorly characterized or have very few known partners. Did you set up any kind of experiment on how this will affect your model?

2. $\textbf{Limited evaluation}$

The main results of the experiments seem to be dependent on moslty one dataset filled with many correlated metrics. Is there any way to expand this to other datasets? How does this dataset [1] perform in your method?


References

PiNUI: A Dataset of Protein-Protein Interactions for Machine Learning; Geoffroy Dubourg-Felonneau and Eyal Akiva and Daniel Wesego and Ranjani Varadan; NeurIPS 2023 Workshop on New Frontiers of AI for Drug Discovery and Development; 2023

**Questions:**

Please refer to the weakness section

---

> ### Author Response · Authors · 2025-12-02
>
> # Reliance on Known Discoveries
> We acknowledge that relying on known discoveries might pose a limitation of our work in cases of poorly characterized proteins. However, it is important to highlight that we do not develop a new model from scratch but we develop a framework that is applied to already trained self-explainable models such as D-SCRIPT and Topsy Turvy. In case a protein is poorly characterized the output would simply be the one of the models. Our contribution is instead to allow ranking candidates in case of known partners, using information that was not previously taken into account by the models.
>
> # Limited Evaluation
> We understand that we only use one dataset for our evaluation. Since other reviewers also raised this point, we have conducted a broader analysis on the suggested PiNUI Dataset [8]. PiNUI is a recently proposed PPI benchmark specifically designed to overcome the well-known biases of classical datasets such as Guo’s and Pan’s. It provides positive interactions from IntAct and employs a principled, nearly uniform negative-sampling strategy that avoids subcellular localization artifacts and forces models to learn true pairwise features rather than dataset-specific shortcuts. Evaluating both D-SCRIPT and our approach on PiNUI, we observe that D-SCRIPT retrieves only a very limited number of novel interactions on PiNUI (Rediscovery Ratio = 0.0080), with extremely low Recall@k and MAP@k, whereas our method achieves a much higher Rediscovery Ratio (0.3849) and consistently better ranking metrics. The higher Avg Rank for our approach reflects the fact that it recovers many more novel partners than D-SCRIPT, which computes its Avg Rank on a much smaller rediscovered subset. Taken together, the results confirm that our approach is not tied to a specific dataset or sampling scheme. Its improvements over D-SCRIPT hold even under PiNUI’s rigorous and bias-controlled conditions, indicating that our method captures robust pairwise interaction signals rather than dataset-specific artifacts.

---

### Author Response · Authors · 2025-12-02

# Summary
We thank the reviewers and the ACs for their efforts.

Our work tackles the problem of retrieving and ranking candidate protein–protein interactions for given target proteins based on their known interaction partners, aiming to prioritize interactions for experimental testing. We propose a novel framework that leverages domain knowledge through interpretability-guided ranking and further refines prioritization by integrating complementary sources of evidence, including interaction scores, structural plausibility, and biomedical language features. We show that our approach yields improvements over state-of-the-art PPI prediction models.

We updated the paper with the suggested changes. To make them clear, we show in red what was removed and in blue what was added.
We provide here a summary of the main points addressed:

 - **Broader evaluation**: We added experiments on the PiNUI benchmark and included xCAPT5 as an additional sequence-based baseline. Our method outperforms probability-only D-SCRIPT and achieves better novel-partner ranking than D-SCRIPT, Topsy-Turvy, and xCAPT5 on STRING v11→v12.

 - **Interpretability-guided retrieval**: We specified better how active residues and regions are extracted from D-SCRIPT/Topsy-Turvy contact maps, emphasizing the automatic selection of the longest activated segment, and clarified that the “cosine method” denotes this interpretability-guided Stage 1 retrieval, not a generic cosine baseline.

 - **Re-ranking and biological signals**: We expanded the description of the four re-ranking signals and clarified that each is applied separately to the top-r candidates, explaining how Table 2 captures directional, non-symmetric rank shifts between methods.

 - **Positioning and novelty**: We clarified that our contribution is not a new prediction model but a novel methodological framework that repurposes existing self-explainable models (D-SCRIPT, Topsy-Turvy) for a prospective ranking task aimed at prioritizing interactions for experimental testing. We show that our approach yields improvements over state-of-the-art PPI prediction models.

 - **Data Leakage**: We clarified that the fine-tuning data used for PubMedBERT is completely disjoint from the evaluation data used in our retrieval experiments. For PubMedBERT, we constructed training pairs exclusively from STRING v11 interaction data. We then applied a GroupKFold split by protein identity, ensuring that all examples involving the same protein were placed in the same fold. This prevents any protein from appearing in both training and validation, and thus avoids leakage at the level of individual proteins. Importantly, the retrieval evaluation uses STRING v12 novel interactions, which are not used in the PubMedBERT fine-tuning process. Therefore, the ranking model never sees during training the specific pairs it is later evaluated on.


All the points are thoroughly addressed in the comments below. We believe we have fully addressed the reviewers' concerns, and we respectfully ask the AC to consider our comments on the points that were raised.

---

> ### Author Response · Authors · 2025-12-02
>
> ### References
> - [1] Altschul, Stephen F., et al. "Basic local alignment search tool." Journal of molecular biology 215.3 (1990): 403-410.
>
> - [2] Eddy, Sean R. "Profile hidden Markov models." Bioinformatics (Oxford, England) 14.9 (1998): 755-763.
>
> - [3] Dang, Thanh Hai, and Tien Anh Vu. "xCAPT5: protein–protein interaction prediction using deep and wide multi-kernel pooling convolutional neural networks with protein language model." BMC bioinformatics 25.1 (2024): 106.
>
> - [4] Li, Xue, et al. "MARPPI: boosting prediction of protein–protein interactions with multi-scale architecture residual network." Briefings in Bioinformatics 24.1 (2023).
>
> - [5] Martin, Shawn, Diana Roe, and Jean-Loup Faulon. "Predicting protein–protein interactions using signature products." Bioinformatics 21.2 (2005): 218-226.
>
> - [6] Guo, Yanzhi, et al. "Using support vector machine combined with auto covariance to predict protein–protein interactions from protein sequences." Nucleic acids research 36.9 (2008): 3025-3030.
>
> - [7] Pan, Xiao-Yong, Ya-Nan Zhang, and Hong-Bin Shen. "Large-Scale prediction of human protein− protein interactions from amino acid sequence based on latent topic features." Journal of proteome research 9.10 (2010): 4992-5001.
>
> - [8] Dubourg-Felonneau, Geoffroy, et al. "PiNUI: A dataset of protein–protein interactions for machine learning." bioRxiv (2023): 2023-12.
>
> - [9] Bryant, Patrick, and Frank Noé. "Rapid protein-protein interaction network creation from multiple sequence alignments with Deep Learning." bioRxiv (2023): 2023-04.
>
> - [10] Devkota, Kapil, James M. Murphy, and Lenore J. Cowen. "GLIDE: combining local methods and diffusion state embeddings to predict missing interactions in biological networks." Bioinformatics 36.Supplement_1 (2020): i464-i473.
>
> - [11] Sledzieski S, Singh R, Cowen L, Berger B. D-script translates genome to phenome with sequence-based, structureaware, genome-scale predictions of protein–protein interactions. Cell Syst. 2021;12(10):969–82.
>
> - [12] Borghini, Alessia, et al. "Identifying Candidates for Protein-Protein Interaction: A Focus on NKp46’s Ligands." CEUR WORKSHOP PROCEEDINGS. Vol. 3831. CEUR-WS, 2024.

---

### Meta-Review · Area_Chair_2iJC · 2026-01-05

**Summary:**

The paper presents a two-stage framework for prioritizing protein-protein interaction (PPI) candidates based on interpretability-guided retrieval and multi-evidence re-ranking. The experimental design is comprehensive, using prospective validation across STRING database versions (v11→v12) and including relevant metrics such as Recall@k and MRR. Despite these positive aspects, fundamental concerns regarding the paper's core contribution remain unresolved:

1) Limited Methodological Novelty: Reviewers consistently noted that the core approach is primarily a post-hoc integration and refinement of existing self-explainable models (D-SCRIPT, Topsy-Turvy). The framework cleverly repurposes their intermediate representations but does not introduce a novel learning paradigm or architectural advance, positioning the work as an incremental rather than foundational contribution.

2) Insufficient Experimental Baselines and Analysis: Key baseline comparisons are still missing, such as classical sequence-similarity methods (BLASTp, HMMER) or a straightforward "translate-then-predict" pipeline, which are necessary to definitively establish the advantage of the proposed interpretability mechanism.
The framework combines multiple biological signals, but a systematic ablation study quantifying the individual contribution of each signal (structural, semantic, LLM-based) is absent. This makes it difficult to assess which components are essential for the observed gains.

3) Practical Limitations and Unclear Generalizability:
The method’s fundamental assumption—that new interactors resemble known ones—inherently limits its applicability to proteins with few or no known partners (orphan proteins), a significant constraint for real-world discovery.
 While cross-dataset results on PiNUI are added, the biological interpretability and practical utility of the top-ranked candidates are not demonstrated through case studies or wet-lab validation, leaving the translational impact unclear.

The paper addresses a relevant problem with a technically sound and practically oriented framework. However, it falls short of the bar for acceptance due to concerns about the degree of methodological innovation, the completeness of the experimental validation, and the scope of its practical applicability. The work is better suited for a venue focused on applications or bioinformatics tools.

**Reviewer Concerns:**

Addressed Reviewer Concerns：
The authors systematically addressed several substantive criticisms in their rebuttal. Key responses include: 1) Supplementing critical experiments, such as adding the PiNUI benchmark and xCAPT5 model comparison, which strengthened the robustness argument for the method; 2) Clarifying core methodological details, explicitly explaining the mechanism for "active residue" extraction and the definition of the "cosine method"; 3) Addressing data leakage concerns by detailing the GroupKFold data splitting strategy based on protein identity used in PubMedBERT fine-tuning, ensuring evaluation used entirely novel interaction data; 4) Improving paper presentation, refining the description of the re-ranking stage and the readability of result tables.

Unresolved Fundamental Issues：
Despite the detailed rebuttal, fundamental disagreements critical to the paper's acceptance remain unresolved: 1) Insufficient innovation of contribution: Multiple reviewers maintain that the work is essentially post-processing and integration of outputs from existing self-explainable models, representing an incremental contribution rather than a methodological breakthrough; this core assessment remained unchanged by the supplementary experiments. 2) Persistent inadequacy in baseline comparisons: Reviewers emphasized the continued lack of direct comparison with classical sequence alignment tools and the absence of a quantitative ablation analysis on the contributions of individual biological evidence sources within the framework, leaving its unique advantages incompletely proven. 3) Inherent limitations of the method: The framework cannot effectively handle "orphan proteins" lacking known interaction partners; this fundamental practical limitation was not resolved, impacting its generalizability.

**Reviewer Scores:**

WiiR (Initial Score: 4)
Projected Final Score: 5
This reviewer's primary concerns were experimental singularity and methodological assumptions. The authors supplemented the PiNUI benchmark experiments and provided a reasonable explanation of the framework's assumptions, offering a direct and effective response.

aVDq (Initial Score: 6)
Projected Final Score: 6-6
This reviewer already held a positive stance, with their extensive comments focusing on in-depth discussion and technical clarification. The authors' detailed responses to all points (e.g., algorithm, runtime, data leakage) will solidify their confidence.

qSmL (Initial Score: 2)
Projected Final Score: 3-4
This reviewer fundamentally questioned the novelty and baseline comparisons. Although the authors supplemented the xCAPT5 baseline and discussion of motivation, they did not add classical sequence alignment baselines or address the core criticism of "incremental contribution." The score may slightly improve due to the supplementary experiments but will likely remain in the borderline/reject range.

X99M (Initial Score: 4)
Projected Final Score: 4
This reviewer's criticism focused on weak innovation and lack of biological analysis. The rebuttal supplemented PiNUI experiments and defended the method but did not conduct ablation studies or provide new biological insights. Since the concern relates to the nature of the contribution, the response is unlikely to change their score.


The rebuttal successfully shifted one borderline reviewer toward support and solidified the position of the positive reviewer. However, it failed to change the perspectives of the two reviewers who questioned the fundamental novelty and scope of the contribution. The division persists, and given the persistent criticism regarding insufficient innovation and missing key baselines, the consensus would likely lean toward a borderline or reject conclusion.

---

### Decision · Program_Chairs · 2026-01-26

Reject